# New Species and Records of *Pleurotheciaceae* from Karst Landscapes in Yunnan Province, China

**DOI:** 10.3390/jof10080516

**Published:** 2024-07-24

**Authors:** Wen-Peng Wang, Darbhe J. Bhat, Lin Yang, Hong-Wei Shen, Zong-Long Luo

**Affiliations:** 1College of Agriculture and Biological Science, Dali University, Dali 671003, China; wwpmicrofungi@163.com (W.-P.W.); yanglin4412@sina.com (L.Y.); hongweifungi@outlook.com (H.-W.S.); 2Department of Botany and Microbiology, College of Science, King Saud University, P.O. Box 2455, Riyadh 11451, Saudi Arabia; bhatdj@gmail.com; 3Center of Excellence in Fungal Research, Mae Fah Luang University, Chiang Rai 57100, Thailand; 4School of Science, Mae Fah Luang University, Chiang Rai 57100, Thailand; 5Cangshan Forest Ecosystem Observation and Research Station of Yunnan Province, Dali University, Dali 671003, China

**Keywords:** morphology, new species, phylogeny, *Savoryellomycetidae*, species diversity

## Abstract

*Pleurotheciaceae* is a genera-rich and highly diverse family of fungi with a worldwide distribution in aquatic and terrestrial habitats. During the investigation of lignicolous freshwater fungi from karst landscapes in Yunnan Province, China, 15 fresh strains were obtained from submerged decaying wood. Based on the morphology and phylogenetic analysis of a combined LSU, ITS, SSU, and *rpb*2 sequence dataset, *Obliquifusoideum triseptatum*, *Phaeoisaria obovata*, *Pleurotheciella brachyspora*, *Pl. longidenticulata*, and *Pl. obliqua* were introduced as new species, *P. synnematica* and *Rhexoacrodictys melanospora* were reported as new habitat records, and *P. sedimenticola* and *Pl. hyalospora* were reported as new collections. In addition, based on morphological comparisons and phylogenetic analysis, we accepted *Obliquifusoideum* into in the family *Pleurotheciaceae* (*Pleurotheciales*, *Savoryellomycetidae*). Freshwater habitats are the primary habitats of *Pleurotheciaceae* species, and Yunnan Province has the highest concentration and species diversity of *Pleurotheciaceae* in China.

## 1. Introduction

*Pleurotheciaceae* was established by Réblová et al. [1] for ten similar genera, *viz. Adelosphaeria*, *Helicoön*, *Monotosporella*, *Melanotrigonum*, *Phaeoisaria*, *Phragmocephala*, *Pleurotheciella*, *Pleurothecium*, *Sterigmatobotrys*, and *Taeniolella*, with *Pleurothecium* as the type genus. Later, Ertz et al. [2] synonymized *Taeniolella* with *Sterigmatobotrys*. Hernández-Restrepo et al. [3] introduced a new genus, *Anapleurothecium* from dead wood. Réblová et al. [4] transferred *Helicoön farinosum* to *Helicoascotaiwania* as *H. farinosum*. Hyde et al. [5] set up a new genus, *Neomonodictys* from decaying wood in Thailand. Based on phylogenetic studies, *Dematipyriforma* and *Rhexoacrodictys* were accepted in *Pleurotheciaceae* [6,7,8,9]. In addition, Dong et al. [6] described two new genera, *Obliquifusoideum* and *Saprodesmium*, and a new species of *Coleodictyospora* from freshwater habitats in China and Thailand. Phylogenetic analyses showed that *Coleodictyospora* and *Saprodesmium* were accepted in *Pleurotheciaceae*, while *Obliquifusoideum* was placed in *Savoryellomycetidae genera incertae sedis*. Recently, Tian et al. [10] introduced *Pseudosaprodesmium* on dead leaves. Thus, presently, sixteen genera, including *Adelosphaeria*, *Anapleurothecium*, *Coleodictyospora*, *Dematipyriforma*, *Helicoascotaiwania*, *Melanotrigonum*, *Monotosporella*, *Neomonodictys*, *Phaeoisaria*, *Phragmocephala*, *Pleurotheciella*, *Pleurothecium*, *Pseudosaprodesmium*, *Rhexoacrodictys*, *Saprodesmium*, and *Sterigmatobotrys*, have been accepted in this family [6,7,8,9,10].

*Obliquifusoideum* was established by Dong et al. [6] to accommodate the type species *O. guttulatum*. The genus is characterized by the following characteristics: superficial, ellipsoidal, black, coriaceous, ostiolate ascomata with a lateral, hyaline-to-dark, subcylindrical neck, and thin peridium, tapering towards the apex, with hypha-like, septate, unbranched paraphyses, unitunicate, eight-spored, cylindrical, short pedicellate asci with a small, refractive, barrel- or jar-shaped apical ring, and fusoid, septate, and hyaline ascospores. Dong et al. [6] placed *Obliquifusoideum* in *Savoryellomycetidae* genera *incertae sedis* due to low support in the phylogenetic analysis. However, the phylogenetic analysis of Jayawardena et al. [9] showed that *Obliquifusoideum* is clustered with other genera of *Pleurotheciaceae* with strong support and, therefore, assigned *Obliquifusoideum* to *Pleurotheciales*.

*Phaeoisaria* was established by Höhnel [11], with *P. bambusae* as the type species. This genus is characterized by long or short erect synnemata with parallelly adpressed conidiophores, a flared fertile above half and apices, polyblastic, sympodially extending the denticulate, with short or long, recurved conidiogenous cells, aseptate or septate, and ellipsoidal or obovoidal conidia [9,12,13]. Luo et al. [14] described the first sexual species, *Phaeoisaria filiformis*, for this genus, and the sexual morph of *Phaeoisaria* share immersed, globose-to-elongate ascomata with a long, cylindrical, black, ostiolar neck, filamentous, branched, septate paraphyses, unitunicate, cylindrical asci with a small refractive apical apparatus, and filiform, multi-septate, hyaline ascospores tapering at both ends.

*Pleurotheciella* was established by Réblová et al. [15] to accommodate two new species from submerged decaying wood, *P. rivularia* and *P. centenaria*, with *P. rivularia* as the type species. The sexual morph of *Pleurotheciella* perithecial, astromatic, semi-immersed-to-superficial, glabrous ascomata, abundant, septate, hyaline paraphyses, unitunicate, cylindric-clavate, eight-spored asci with an apical ring, and fusoid-to-fusiform, hyaline, septate ascospores. The asexual morph of *Pleurotheciella* characteristics by reduced or cylindrical, hyaline or brown, septate, unbranched conidiophores, terminal or integrated, subhyaline-to-hyaline, sympodially denticulate conidiogenous cells, and ellipsoidal-to-obovoidal or muriform, variedly shaped, unicellular or septate conidia [7,12,15,16]. Sixteen species have so far been accepted in *Pleurotheciella*, with worldwide distribution including aquatic habitats, except *P. dimorphospora* [1,4,7,12,15,16,17,18].

*Rhexoacrodictys* was established by Baker et al. [19] to accommodate the type species, *R. erecta*. Eight epithets are listed in the Index Fungroum [20]. Based on phylogenetic analyses, *Rhexoacrodictys martini*, *R. nigrospora*, and *R. queenslandica* were transferred to *Distoseptispora*, *Dematipyriforma*, and *Junewangia*, respectively [8,21], and *R. broussonetiae* and *R. fuliginosa* were subsequently added to the genus based on morphological studies [22,23]. Currently, phylogenetic studies on only three species of *Rhexoacrodictys* (*R. erecta*, *R. fimicola*, and *R. melanospora*) [24] are known.

Karst landforms are one of the most typical landforms in the world [25,26]. In China, karst landforms are mainly distributed in the southwest areas, especially the Guangxi Zhuang Autonomous Region, Guizhou Province, and Yunnan Province [27]. Yunnan Province is the most studied area of lignicolous freshwater fungi in China [14,28,29,30], but there is no report on lignicolous freshwater fungi in karst landscapes. In this study, 15 fresh fungal strains were found on submerged decaying wood from karst landscapes in Yunnan Province, China. Based on combined multi-loci phylogenetic analysis and morphological characteristics, nine species were identified, including five new species, two new habitat records, and two new collections. Furthermore, we transferred *Obliquifusoideum* to *Pleurotheciaceae* (*Pleurotheciales*, *Savoryellomycetidae*).

## 2. Materials and Methods

### 2.1. Sample Collection

Specimens of submerged decaying wood were collected from freshwater streams in karst landscapes in Honghe Hani and Yi Autonomous Prefecture, Qujing City, and Wenshan Zhuang and Miao Autonomous Prefecture, Yunnan Province, China, during the dry season (February 2023) and wet season (July 2023). To preserve their integrity, the specimens were transported to the laboratory in plastic bags. The sample processing was by described Shen et al. [31]: the samples were cut to the appropriate length, numbered, and placed in a disinfected plastic crisper for incubated culture at room temperature.

### 2.2. Isolation and Morphological Examination

Fungal colonies on natural substrates were observed using a Guiguang GL-99BI compound stereomicroscope (Guilin Guiguang Instrument Co., Ltd., Guilin, China) and then photographed with a Nikon SMZ1000 stereo zoom microscope (NIKON CORPORATION, Tokyo, Japan). Fungal structures were photographed using a Nikon ECLIPSE Ni-U compound microscope (NIKON CORPORATION, Tokyo, Japan) fitted with a Nikon DS-Ri2 digital camera (NIKON CORPORATION, Tokyo, Japan), as per the guidelines provided by Luo et al. [32] and Senanayake et al. [33]. Single spore isolation was conducted by following the methods described by Shen et al. [31]. Measurements were made with the Tarosoft (R) Image Frame Work program, and photo plates representing fungal structures were processed in Adobe Photoshop CS5 software (Adobe Systems Inc., San Jose, CA, USA). Herbarium specimens (dry woody branches with fungal material) were deposited in the herbarium of Cryptogams, Kunming Institute of Botany Academia Sinica (KUN-HKAS), Kunming, China. The isolates obtained in this study were deposited in the China General Microbiological Culture Collection Center (CGMCC), Beijing, China, and the Kunming Institute of Botany Culture Collection Center (KUNCC), Kunming, China. Names of the new taxa were registered in Fungal Names (FN) (https://nmdc.cn/fungalnames/, accessed on 14 June 2024).

### 2.3. Isolation and Morphological Examination

A Trelief^TM^ Hi-Pure Plant Genomic DNA Kit (Beijing TsingKe Biotech Co., Ltd., Beijing, China) was used to extract total genomic DNA from fungal mycelia. DNA amplification was performed by a polymerase chain reaction (PCR). Four partial gene regions, the large subunit of the nuclear ribosomal RNA gene (LSU), the nuclear ribosomal internal transcribed spacer (ITS), the small subunit of the nuclear ribosomal RNA gene (SSU), and the second-largest subunit of RNA polymerase II (*rpb*2), were used in this study. Sequences of LSU, ITS, SSU, and *rpb*2 were amplified using primer pairs LR0R/LR5, ITS5/ITS4, NS1/NS4, and fRPB2-5F/fRPB2-7cR, respectively [34,35,36]. The amplification was performed in a 25 µL reaction volume containing 9.5 µL of deionized water, a 12.5 µL 2 × Taq PCR Master Mix with blue dye (Sangon Biotech, Shanghai, China), 1 µL of DNA template and 1 µL of each primer (10 µm). The PCR thermal cycling conditions of ITS and SSU were as follows: 94 °C for 3 min, followed by 35 cycles of denaturation at 94 °C for 30 s, annealing at 56 °C for 50 s, elongation at 72 °C for 1 min, and a final extension at 72 °C for 10 min. The LSU thermal cycling conditions were as follows: 94 °C for 3 min, followed by 35 cycles of denaturation at 94 °C for 30 s, annealing at 55 °C for 50 s, elongation at 72 °C for 1 min, and a final extension at 72 °C for 10 min. The *rpb2* has a total of 40 cycles, and the conditions are as follows: initial denature at 95 °C for 5 min before entering 40 cycles; then, denaturation occurs at 95 °C for 1 min, annealing at 52 °C for 2 min, extension at 72 °C for 90 s, and finally at 72 °C for 10 min. PCR products were checked on 1% agarose electrophoresis gels stained with Gel Red. The sequencing reactions were carried out with the primers mentioned above by Tsingke Biological Engineering Technology and Services Company, Kunming, China.

### 2.4. Phylogenetic Analyses

BLAST searches were performed to find similar sequences that matched our data. The sequences were aligned using the online multiple alignment program MAFFT version 7 [37], and this alignment was manually optimized in BioEdit v.7.0.5.3 [38]. The single-gene dataset was concatenated by SquenceMatrix v.1.7.8 for multi-gene phylogenetic analyses [39]. The alignment formats were changed to PHYLIP and NEXUS formats by the AliView and ALigment Transformation EnviRonment (ALTER) website (http://sing.ei.uvigo.es/ALTER/, accessed on 14 June 2024).

Maximum likelihood (ML) analysis was performed using RAxML-HPC2 on ACCESS [40,41] on the CIPRES Science Gateway website ([42]: http://www.phylo.org/portal2, accessed on 14 June 2024), and the estimated proportion of invariant sites were (GTRGAMMA+I) modeled. Bayesian analysis was performed in MrBayes 3.2.6 [43], and the best-fit model of sequence evolution was estimated via MrModeltest 2.2 [44,45,46]. The Markov Chain Monte Carlo (MCMC) sampling approach was used to calculate posterior probabilities (PP) [47]. Bayesian analysis of six simultaneous Markov chains was run for 10,000,000 generations, with trees sampled every 1000 generations.

Phylogenetic trees were visualized using FigTree v. 1.4.0 ([48]: http://tree.bio.ed.ac.uk/software/figtree/, accessed on 14 June 2024), edited in Microsoft Office PowerPoint. Sequences generated in this study were deposited in GenBank and are listed in Table 1.

## 3. Results

### 3.1. Phylogenetic Analyses

The dataset of combined LSU, ITS, SSU, and *rpb*2 sequence data comprises 129 strains with 3400 characters, including gaps (LSU: 1–791 bp, ITS: 792–1318 bp, SSU: 1319–2255 bp, and *rpb*2: 2256–3400 bp). *Conioscypha lignicola* (CBS 335.93) and *C. minutispora* (CBS 137253), were selected as the outgroup taxa. RAxML and Bayesian analyses were conducted and resulted in generally congruent topologies. The best RAxML tree with a final likelihood value of –34,142.771953 is presented. The matrix had 1670 distinct alignment patterns, with 33.53% undetermined characters or gaps. Estimated base frequencies were as follows: A = 0.234926, C = 0.259159, G = 0.291032, T = 0.214883; substitution rates AC = 1.436665, AG = 3.055655, AT = 1.516984, CG = 1.101298, CT = 7.087094, and GT = 1.000000; the gamma distribution shape parameter was α = 0.224395.

In the phylogenetic tree, fifteen newly obtained strains were nested in *Obliquifusoideum*, *Phaeoisaria*, *Pleurotheciella*, and *Rhexoacrodictys* (Figure 1). *Obliquifusoideum triseptatum* (CGMCC 3.27014) clusters in *Obliquifusoideum* and is sister to *O. guttulatum* (MFLUCC 18–1233) with 100% ML/1.00 PP support; *Phaeoisaria obovata* (CGMCC 3.27015 and KUNCC 23–15598) was sister to *P. aquatica* (MFLUCC 16–1298) and *P. siamensis* (MFLUCC 16–0607) with 0.99 PP support; *Pleurotheciella brachyspora* (CGMCC 3.25435), *Pl. longidenticulata* (CGMCC 3.27018), and *Pl. obliqua* (CGMCC 3.27019 and KUNCC 23–16569) were clustered with *Pl. dimorphospora* (KUMCC 20–0185) and *Pl. saprophytica* (MFLUCC 16–1251) and formed a separate clade for *Pleurotheciella*, *Phaeoisaria sedimenticola* (KUNCC 23–14648 and KUNCC 23–15613); *P. synnematica* (KUNCC 23–16573 and KUNCC 23–16619), *Pl. hyalospra* (CGMCC 3.27017, KUNCC 23–16648, KUNCC 23–16664), and *Rhexoacrodictys melanospora* (KUNCC 23–16529) were clustered with *P. sedimenticola* (CGMCC 3.14949, KUNCC 10456, S-908), *P. synnematica* (NFCCI 4479), *Pl. hyalospra* (GZCC 22–2018 and GZCC 22–2023), and *R. melanospora* (KUNCC 22–12406 and KUNCC 22–12411), respectively (Figure 1).

### 3.2. Taxonomy

***Obliquifusoideum triseptatum*** W.P. Wang, H.W. Shen & Z.L. Luo, sp. nov., Figure 2.

Fungal Names number: FN 571959.

Etymology: Refers to the ascospores with three septa.

Holotype: HKAS 131970

*Saprobic* on submerged decaying wood. **Asexual morph**: Undetermined. **Sexual morph**: *Ascomata* 170–190 µm high, 210–270 µm diam., scattered, superficial to semi-immersed, oval to subglobose, black, ostiolate, with a short, subcylindrical, mostly perpendicular neck. *Ostiole* periphysate. *Peridium* 16–32 µm thick, composed of thin-walled, green, nearly long rectangle cells of *textura angularis* in the outer layers, becoming hyaline, oval or irregular cells with small guttulate of *textura angularis* towards inner layers. *Paraphyses* 2.2–4.7 µm wide, tapering towards the apex, septate, branched, hyaline, embedded in a gelatinous matrix. *Asci* (90–) 110–160 × 5.9–13 µm ($\bar{x}$ = 125.5 × 8.8 µm, n = 30), 8-spored, unitunicate, cylindrical, slightly flexuous, rounded at the apex, (7.5–) 13–18 µm long pedicellate, with an apical ring. *Ascospores* 13–29 × 4–7.3 µm ($\bar{x}$ = 20.8 × 5.8 µm, n = 40), overlapping uniseriate, fusoid to fusiform, mostly dull at both ends, straight or slightly curved, one median septate, with two additional obscure septa at both sides, occasionally slightly constricted at the central septum, guttulate, hyaline, thin and smooth-walled, without a gelatinous sheath.

Culture characteristics: Ascospores germinate on the potato dextrose agar (PDA) within 24 h, with germ tubes produced from both ends. Colonies growing on PDA after 3 weeks of incubation at room temperature attain a diameter of about 15 mm. Mycelia dry and dense. Colonies form on the surface of PDA, with regular edges and a rough surface, khaki-to-brown. The reverse is khaki-to-brown and smooth.

Material examined: China, Yunnan Province, Wenshan Zhuang, and Miao Autonomous Prefecture, Qiubei County (24°15′59.24″ N; 104°09′19.76″ E), on submerged decaying wood, 17 July 2023, Wen-Peng Wang, S-5725 (HKAS 131970, holotype), ex-type culture CGMCC 3.27014 = KUNCC 23–16650.

Notes: Phylogenetic analysis showed that *Obliquifusoideum triseptatum* (CGMCC 3.27014) clustered in *Obliquifusoideum* and is sister to *O. guttulatum* (MFLUCC 18–1233) with 100% ML/1.00 PP support (Figure 1). Morphologically, *O. triseptatum* resembles *O. guttulatum* with cylindrical, short pedicellate asci with a barrel- or jar-shaped apical ring and overlapping uniseriate, fusoid ascospores. However, *O. triseptatum* has superficial-to-semi-immersed, oval-to-subglobose ascomata with a short, perpendicular neck, asci with rounded apical, and larger ascospores (13–29 × 4–7.3 vs. 14–17.5 × 4.3–5 µm) different from *O. guttulatum* [6]. We, therefore, introduce *O. triseptatum* as a new species in *Obliquifusoideum*.

Based on phylogenetic analysis, Dong et al. [6] established the genus *Obliquifusoideum* in *Savoryellomycetidae* genera *incertae sedis*. Several phylogenetic studies have yielded the same results, where *Obliquifusoideum* constitutes an independent lineage that is basal to other genera of *Pleurotheciaceae* [6,8,9] (Figure 1). We compared the morphology of *Obliquifusoideum* with genera in *Pleurotheciaceae. Obliquifusoideum* has semi-immersed-to-superficial, subglobose ascomata, thin peridium, abundant, septate paraphyses, eight-spored, cylindrical asci with an apical ring, and three-septate, fusoid to fusiform ascospores, which are similar to *Pleurotheciaceae* [1,4,6,12,15]. Therefore, we accommodated *Obliquifusoideum* in *Pleurotheciaceae* (*Pleurotheciales*, *Savoryellomycetidae*).

***Phaeoisaria obovata*** W.P. Wang, H.W. Shen & Z.L. Luo, sp. nov., Figure 3.

Fungal Names number: FN 571975.

Etymology: Referring to the obovoid conidia.

Holotype: HKAS 13198.3

*Saprobic* on submerged decaying wood. **Asexual morph**: *Colonies* effuse, solitary, dark brown to black, hairy, covered by white conidial. *Mycelium* partly immersed, partly superficial, composed of septate, branched, pale brown hyphae. *Synnemata* 150–1370 × 7.1–21 µm ($\bar{x}$ = 727.5 × 14.6 µm, n = 10), scattered, erect, rigid, dark brown to black, sometimes flared at the apex, pale at the apex, composed of compactly adpressed conidiophores. *Conidiophores* macronematous, synnematous, septate, cylindrical, unbranched to branched, straight, dark brown, paler at the apex, smooth-walled. *Conidiogenous cells* 7.6–22 × (1.2–) 1.5–2.3 (–2.8) µm ($\bar{x}$ = 14.3 × 2 µm, n = 20), integrated, terminal and intercalary, polyblastic, curved to recurved, longer at the apex of synnema, smooth at the base, cylindrical or tapering towards the tip, subhyaline to pale brown, with several small denticulate conidiogenous loci. *Secession* schizolytic. *Conidia* 3.7–7.9× 2–3.2 µm ($\bar{x}$ = 5.6 × 2.5 µm, n = 40), solitary, subglobose to obovoidal to elongated obovoidal, smooth to finely verrucose, rounded apical and obtuse basal, hyaline, aseptate, straight. **Sexual morph**: Undetermined.

Culture characteristics: Conidia germinate on PDA within 24 h, and germ tubes are produced from both ends. Colonies on PDA after 4 weeks of incubation at room temperature attain a diameter of about 2 cm. Mycelia dry and dense. Colonies on the surface of PDA are raised, with irregular edges, brittle, rough at the surface, and brown-to-dark brown. The reverse is brown-to-dark brown, lighter at the edges, and smooth.

Material examined: China, Yunnan Province, Honghe Hani and Yi Autonomous Prefecture, Mile City (24°42′69.75″ N; 103°48′34.68″ E), on submerged decaying wood, 14 July 2023, Wen-Peng Wang, S-5352 (HKAS 131983, holotype), ex-type culture CGMCC 3.27015 = KUNCC 23–15595; Qujing City, Luoping County (25°01′52.57″ N; 104°42′47.40″ E), on submerged decaying wood, 15 July 2023, Zheng-Quan Zhang, S-5356 (HKAS 131966, paratype), living culture KUNCC 23–15598.

Notes: Phylogenetic analysis showed that *Phaeoisaria obovata* (CGMCC 3.27015 and KUNCC 23–15598) is close to *P. aquatica* (MFLUCC 16–1298), *P. guttulata* (MFLUCC 17–1965) and *P. siamensis* (MFLUCC 16–0607) (Figure 1). Morphologically, *P. obovata* differs from *P. aquatica* in having longer synnemata (150–1370 vs. 313–727 µm) and smooth-to-finely verrucose conidia [12,16]; *P. obovata* has recurved, longer conidiogenous cells (7.6–22 vs. 8–12 µm), and obovoidal-to-elongated obovoidal conidia which differs from *P. siamensis* [49]; and *P. obovata* is distinguished from *P. guttulata* by longer conidia (3.7–7.9 vs. 3.5–5.5 µm) with obtuse basal [17]. We, therefore, introduce *Phaeoisaria obovata* as a new species.

***Phaeoisaria sedimenticola*** X.L. Cheng & Wei Li ter, Mycotaxon 127 (1): 20 (2014), Figure 4.

Fungal Names number: FN563661.

*Saprobic* on submerged decaying wood. **Asexual morph**: *Colonies* effuse, solitary, dark brown to black, hairy, covered by white conidial. *Mycelium* partly immersed, partly superficial, composed of septate, branched, brown hyphae. *Synnemata* 560–1510 × 14–42 µm ($\bar{x}$ = 1051.9 × 26 µm, n = 10), solitary or gathered, erect, rigid, subulate, tapering towards the apex, pale brown to black, paler towards the apex, composed of compact appressed conidiophores. *Conidiophores* macronematous, synnematous, septate, cylindrical, branched, straight, pale brown to black, pale pigment at the apex, smooth-walled. *Conidiogenous cells* 14–31 ×1.6–2.4 µm ($\bar{x}$ = 21.2 × 2 µm, n = 20), integrated, terminal and intercalary, polyblastic, fertile portion bent outwards, smooth-walled, with multiple small, hyaline, cylindrical denticulate conidiogenous loci clustered in the apical part. *Conidia* 6.3–10.2× 2.5–3.6 µm ($\bar{x}$ = 8.2 × 3.1 µm, n = 40), solitary, obovoid to subcylindrical, smooth, rounded apical and obtuse basal, hyaline, aseptate, straight, guttulate. **Sexual morph**: Undetermined.

Culture characteristics: Conidia are germinated on PDA within 24 h, and germ tubes are produced from both ends. Colonies are obtained on PDA after 4 weeks of incubation at room temperature, attaining a diameter of about 15 mm. Mycelia dry and dense. Colonies on the surface of PDA are brown, raised, with regular edges, and brittle and rough at the surface. The reverse is brown-to-dark brown, lighter at the edges, and smooth.

Material examined: China, Yunnan Province, Qujing City, Luoping County (24°92′51.39″ N; 104°28′78.54″ E), on submerged decaying wood, 15 July 2023, Wen-Peng Wang, S-5128 (HKAS 131978), living culture KUNCC 23–14648; Qujing City, Luoping County (24°76′57.73″ N; 104°49′72.38″ E), on submerged decaying wood, 16 July 2023, Wen-Peng Wang, S-5382 (HKAS 131971), living culture KUNCC 23–15613.

Notes: *Phaeoisaria sedimenticola* was introduced by Cheng et al. [50] and isolated from the surface marine sediment in Weihai, Shandong Province, China. Afterwards, Xu et al. [51] recorded this species from freshwater habitat in Xizang, China. In the phylogenetic analysis, our new collections (KUNCC 23–14648 and KUNCC 23–15613) clustered with three strains of *P. sedimenticola* (CGMCC 3.14949, S-908, KUNCC 10456) (Figure 1). Morphologically, our new collections fit well with the description of *P. sedimenticola* in having subulate synnemata, a fertile portion when bent outwards, cylindrical denticulate conidiogenous cells, and smooth-walled, obovoidal, aseptate conidia. We, therefore, identify our new collections as *P. sedimenticola*.

***Phaeoisaria synnematica*** P.N. Singh & S.K. Singh, Fungal Diversity 111: 191 (2021), Figure 5.

Fungal Names number: FN830711.

*Saprobic* on submerged decaying wood. **Asexual morph**: *Colonies* effuse, scattered or the difference in length are significant gathered in small group, dark brown to black, hairy, covered by white, velvety conidial. *Mycelium* partly immersed, partly superficial, composed of septate, branched, brown hyphae. *Synnemata* 120–660 × 3.1–10.6 (–17) µm ($\bar{x}$ = 325.6 × 8.2 µm, n = 15), erect, rigid, dark brown to black, composed of compact appressed conidiophores. *Conidiophores* macronematous, synnematous, cylindrical, septate, branched, straight, dark brown, smooth-walled. *Conidiogenous cells* 8.2–31 (–61) × 1.6–2.8 µm ($\bar{x}$ = 21.5 × 2.2 µm, n = 20), integrated, terminal and intercalary, polyblastic, cylindrical to subcylindrical, recurved, rarely branched, with several small denticulate, subhyaline to brown conidiogenous loci. *Secession* schizolytic. *Conidia* (3.5) 4.8–6.7× 2.2–3.2 µm ($\bar{x}$ = 5.7 × 2.5 µm, n = 40), solitary, obovoidal, smooth-walled, rounded apical and obtuse basal, hyaline, aseptate, straight, small guttulate. **Sexual morph**: Undetermined.

Culture characteristics: Conidia germinating on PDA within 24 h, and germ tubes produced from both ends. Colonies growing on PDA, and after 4 weeks of incubation at room temperature, attaining a diameter of about 18 mm. Mycelium dry, dense. Colonies semi-immersed in PDA, pie-like protrusion in central, with irregular edges, brittle, rough surface, dark brown and covered the surface with a layer of gray. Reverse brown to dark green, lighter center and edges, smooth. *Synnemata* 443.4–1429 µm long, erect, rigid, cylindrical, dark brown to black, composed of incompact conidiophores. *Conidiophores* macronematous, synematous, cylindrical, septate, branched, slightly flexure, brown. *Conidiogenous cells* 10.6–31.1 (–45.7) × 1.7–2.7 µm ($\bar{x}$ = 22.2 × 2.2 µm, n = 20), integrated, terminal and intercalary, monoblastic or polyblastic, subcylindrical, recurved, unbranched, without or with multiple small denticulate conidiogenous loci, hyaline to brown. *Conidia* 4–7.4 × 2.4–3.6 µm ($\bar{x}$ = 6 × 3 µm, n = 40), solitary, obovoidal, thin, smooth-walled, rounded at the apex, hyaline, aseptate, slightly truncate at the base. *Chlamydospores* 6.9–13.2 × 5.3–9.7 µm ($\bar{x}$ = 9.8 × 7.2 µm, n = 30), produced from conidiophores, intercalary, lateral to terminal, solitary or occasionally catenate, obovoidal or globose to subglobose, thick, smooth-walled, rounded at the apex, guttulate, aseptate, hyaline when young, dark brown when mature. *Secession* schizolytic.

Material examined: China, Yunnan Province, Qujing City, Luoping County (24°76′57.73″ N; 103°49′72.38″ E), on submerged decaying wood, 16 July 2023, Xing-Ya Zeng, S-5544 (HKAS 131965), living culture KUNCC 23–16573; Qujing City, Luoping County (24°74′13.01″ N; 104°47′05.92″ E), on submerged decaying wood, 16 July 2023, Wen-Peng Wang, S-5657 (HKAS 131974), living culture KUNCC 23–16619.

Notes: Phylogenetic analysis showed that our new collections (KUNCC 23–16573 and KUNCC 23–16619) were clustered with the type strains of *Phaeoisaria synnematica* (NFCCI 4479) (Figure 1). Our new collections are similar to *P. synnematica* with branched conidiophores, terminal or intercalary, cylindrical, branched conidiogenous cells, and intercalary, lateral-to-terminal, obovoidal or globose-to-subglobose chlamydospores [16]. *Phaeoisaria synnematica* was introduced by Boonmee et al. [16] and collected from the dead bark of *Azadirachta indica* (*Meliaceae*) in India, however our two new collections were collected from freshwater habitats. Therefore, we identified our new collections as *Phaeoisaria synnematica* based on morphological and phylogenetic analysis, and it is the first time to report this species from freshwater habitats in China.

***Pleurotheciella brachyspora*** W.P. Wang, H.W. Shen & Z.L. Luo, sp. nov., Figure 6.

Fungal Names number: FN 571976.

Etymology: Referring to the short conidia.

Holotype: HKAS 131981.

*Saprobic* on submerged decaying wood. **Asexual morph**: *Colonies* superficial, effuse, mostly gathered in small groups, partly scattered, hairy, silvery to dark brown, glistening. *Mycelium* mostly immersed, composed of septate, smooth-walled, unbranched, pale brown hyphae. *Conidiophores* 57–92 × 2.5–4.2 µm ($\bar{x}$ = 75.8 × 3.3 µm, n = 20), macronematous, mononematous, cylindrical, straight or slightly flexuous, solitary or grouped, unbranched, septate, brown, paler towards the apex, entirely fertile, covered by denticulate conidiogenous cells except the base. *Conidiogenous cells* polyblastic, integrated, terminal and intercalary, up to 1.6 µm long, cylindrical, subhyaline to pale brown denticulate. *Conidia* (5.9–) 8.2–11 × 2.8–3.7 µm ($\bar{x}$ = 9.4 × 3.3 µm, n = 30), acrogenous, solitary, subhyaline, guttulate, straight or slightly curved, obovoid to fusoid, smooth, rounded at the apex, obtuse or tapering towards the base, aseptate when young, 1-septate at maturity. **Sexual morph**: Undetermined.

Culture characteristics: Conidia germinate on PDA within 24 h, with germ tubes produced from the apex. Colonies growing on PDA after 3 weeks of incubation at room temperature attain a diameter of about 15 mm. Mycelia dry and dense, with colonies semi-immersed in PDA, with regular edges, a gray, rough surface, and grid texture. The reverse is dark green and smooth.

Material examined: China, Yunnan Province, Wenshan Zhuang and Miao Autonomous Prefecture, Guangnan County (24°29′33.74″ N; 105°07′02.35″ E), on submerged decaying wood, 28 February 2023, Wen-Peng Wang S-4362 (HKAS 131981, holotype), ex-type culture CGMCC 3.25435 = KUNCC 23–13753.

Notes: Phylogenetic analysis showed that *Pleurotheciella brachyspora* (CGMCC 3.25435) constitutes an independent lineage that is basal to *P. longidenticulata* (CGMCC 3.27018), *P. obliqua* (CGMCC 3.27019 and KUNCC 23–16569), and *P. saprophytica* (MFLLUCC 16–1215) with 93% ML/1.00 PP support (Figure 1). Morphologically, all of these species have denticulate conidiogenous cells covered to the middle of conidiophores from the tip, whereas *P. brachyspora* differs from *P. longidenticulata* in having cylindrical conidiophores, shorter denticles (1.6 vs. 3.7 µm); *P. brachyspora* differs from *P. saprophytica* in having longer conidiophores (57–92 vs. 44–52 µm) and cylindrical conidiogenous cells [12]; and *P. brachyspora* sometimes has tapered basal and aseptate conidia, different from *P. obliqua*. We, therefore, introduce *Pleurotheciella brachyspora* as a new species.

***Pleurotheciella hyalospora*** J. Ma & Y.Z. Lu, Fungal Diversity 124: 61 (2024), Figure 7.

Fungal Names number: FN 900171.

*Saprobic* on submerged decaying wood. **Asexual morph**: *Colonies* on the substratum superficial, effuse, often in small groups, partly scattered, hairy, silvery to brown, glistening. *Mycelium* mostly immersed, composed of septate, smooth-walled, unbranched, pale brown hyphae. *Conidiophores* 43–130 × 2.2–3.2 µm ($\bar{x}$ = 77.5 × 2.7 µm, n = 30), macronematous, mononematous, cylindrical, straight or slightly flexuous, unbranched, septate, brown, paler towards the apex, hyaline at the apex, with small denticulate conidiogenous cells at the apex. *Conidiogenous cells* polyblastic, integrated, terminal, up to 1.2 µm long, cylindrical, subhyaline denticulate. *Conidia* 12–18 × 2.8–4.1 µm ($\bar{x}$ = 15 × 3.5 µm, n = 40), acrogenous, solitary, subhyaline, guttulate, straight or curved, fusoid to clavate, rounded apical and obtuse basal, smooth-walled, aseptate when young, uniseptate at maturity. **Sexual morph**: Undetermined.

Culture characteristics: Conidia germinate on PDA within 24 h, with germ tubes produced from the apex. Colonies grow on PDA after 4 weeks of incubation at room temperature, attaining a diameter of about 2 cm. Mycelia dry and dense. Colonies form on the surface of PDA, with irregular edges, rough, and have a khaki-to-dark brown color. The reverse is khaki, smooth, and sectored from the center.

Material examined: China, Yunnan Province, Qujing City, Luoping County (24°96′36.10″ N; 104°29′01.27″ E), on submerged decaying wood, 15 July 2023, Wen-Peng Wang, S-5473 (HKAS 131975), living culture CGMCC 3.27017 = KUNCC 23–15669; Wenshan Zhuang and Miao Autonomous Prefecture, Guangnan County (24°29′33.74″ N; 105°07′02.35″ E), on submerged decaying wood, 20 July 2023, Fa-Li Li, S-5721 (HKAS 131979, paratype), living culture KUNCC 23–16648; Qujing City, Luoping County (24°92′51.39″ N; 104°28′78.54″ E), on submerged decaying wood, 15 July 2023, Wen-Peng Wang, S-5745 (HKAS 131982), living culture KUNCC 23–16664.

Notes: Phylogenetic analysis showed that our new strains (CGMCC 3.27017, KUNCC 23–16648, KUNCC 23–16664) clustered with two strains of *Pleurotheciella hyalospora* (GZCC 22–2018 and GZCC 22–2023) (Figure 1). Morphologically, our new collections are similar to *P. hyalospora* with unbranched, straight or slightly flexuous, cylindrical conidiophores, terminal conidiogenous cells with denticles, and fusoid-to-clavate, curved, uniseptate conidia with rounded apical and obtuse basal. *P. hyalospora* was introduced by Liu et al. [18] from freshwater and terrestrial habitats in Guizhou Province, China. Therefore, we identified our new collections as *Pleurotheciella hyalospora* based on morphological and phylogenetic analysis.

***Pleurotheciella longidenticulata*** W.P. Wang, H.W. Shen & Z.L. Luo, sp. nov., Figure 8.

Fungal Names number: FN 571977.

Etymology: Referring to conidiogenous cells with longer denticulate.

Holotype: HKAS 131973.

*Saprobic* on submerged decaying wood. **Asexual morph**: *Colonies* superficial, effuse, mostly in small groups, sometimes scattered, hairy, subhyaline to brown. *Mycelium* mostly immersed, composed of septate, smooth-walled, unbranched, pale brown hyphae. *Conidiophores* (26–) 47–100 (–123) × 2.2–3.2 µm ($\bar{x}$ = 67.2 × 2.7 µm, n = 30), macronematous, mononematous, solitary or grouped, subcylindrical, with irregular expansion and constricted, straight or slightly flexuous, septate, unbranched, brown at the base, paler towards the apex, except for the base covered denticulate conidiogenous cells. *Conidiogenous cells* polyblastic, integrated, terminal and intercalary, up to 3.7 µm long, cylindrical, subhyaline to pale brown to brown denticulate. *Conidia* 7.2–12 × (2.1–) 2.6–3.9 µm ($\bar{x}$ = 9.4 × 3.1 µm, n = 40), acrogenous, solitary, subhyaline, guttulate, straight, cylindrical to ellipsoid, obtuse at both ends, smooth-walled, 1-septate, with an inconspicuous central septum. **Sexual morph**: Undetermined.

Culture characteristics: Conidia germinate on PDA within 24 h, with germ tubes produced from both ends. Colonies grow on PDA after 4 weeks of incubation at room temperature, attaining a diameter of about 15 mm. Mycelia dry and dense. Colonies on the surface of PDA are raised, with regular edges, a rough surface, brittle, and white-to-brown. The reverse is brown-to-dark brown, with white edges, smooth, and sectored from the center.

Material examined: China, Yunnan Province, Qujing City, Luoping County (24°74′13.01″ N; 104°47′05.92″ E), on submerged decaying wood, 16 July 2023, Wen-Peng Wang, S-5510 (HKAS 131973, holotype), ex-type culture CGMCC 3.27018 = KUNCC 23–15692.

Notes: *Pleurotheciella longidenticulata* resembles *P. obliqua* with solitary or grouped conidiophores, which are paler towards at the apex, covered with integrated, terminal and intercalary, denticulate, cylindrical conidiogenous cells, and straight, obovoid-to-cylindrical, 1-septate conidia. However, the phylogenetic analysis showed that *P. longidenticulata* (CGMCC 3.27018) is sister to *P. saprophytica* (MFLUCC 16–1251) with 100% ML/1.00 PP support (Figure 1). Morphologically, *Pleurotheciella longidenticulata* has solitary or grouped, longer ((26–) 47–100 (–123) vs. 44–52 µm), subcylindrical conidiophores with irregular expansion and contraction, cylindrical denticulate conidiogenous cells, and cylindrical-to-ellipsoid conidia are obtuse at both ends, which is different from *P. saprophytica* [12]. A comparison of the ITS, LSU, and *rpb*2 sequences between *P. longidenticulata* and *P. saprophytica* showed 1.57% (8/508 bp, 4 gaps), 0.38% (3/780 bp), and 3.20% (25/781 bp) differences, respectively. Therefore, based on morphological and molecular sequence evidence, we introduce *Pleurotheciella longidenticulata* as a new species.

***Pleurotheciella obliqua*** W.P. Wang, H.W. Shen & Z.L. Luo, sp. nov., Figure 9.

Fungal Names number: FN 571978.

Etymology: Referring to the sexual morph of this fungal has an oblique growing the neck.

Holotype: HKAS 131980.

*Saprobic* on submerged decaying wood. **Asexual morph**: *Colonies* superficial, effuse, most in small groups, sometimes scattered, hairy, subhyaline to dark brown. *Mycelium* mostly immersed, composed of septate, smooth-walled, unbranched, pale brown hyphae. *Conidiophores* 32–100 × 2.2–3.3 µm ($\bar{x}$ = 61.5 × 2.6 µm, n = 30), macronematous, mononematous, solitary or gathered, subcylindrical, straight or flexuous, septate, unbranched, brown, paler towards the apex, subhyaline at the apex, covered, with denticulate conidiogenous cells except the base. *Conidiogenous cells* polyblastic, integrated, terminal and intercalary, up to 2.2 µm long denticulate, cylindrical, subhyaline to pale brown to brown. *Conidia* 7.1–12 × 2.3–3.5 µm ($\bar{x}$ = 10.2 × 2.9 µm, n = 40), acrogenous, solitary, hyaline, guttulate, straight, clavate to subcylindrical, rounded apical and obtuse basal, smooth-walled, 1-septate. **Sexual morph**: *Ascomata* 75–130 µm high, 91–160 µm diam., scattered or aggregated, superficial, prostrate, subellipsoidal, dark brown to black, ostiolate, with a lateral neck. *Neck* 96–160 µm long., subhyaline to black, subcylindrical, first horizontal, then bent oblique to the substrate. *Peridium* 7.4–30 µm thick, coriaceous, composed 5–7 layers of pale brown to brown, irregular polyhedral, thin-walled cells of *textura angularis. Paraphyses* 2.4–4.2 (–7.1) µm wide, short, subcylindrical, dense, septate, sometimes constricted at the septum, unbranched, hyaline, embedded in a gelatinous matrix. *Asci* 58–95 (–120) × 5.5–11 µm ($\bar{x}$ = 85.2 × 7.8 µm, n = 25), 8-spored, unitunicate, cylindrical to clavate, slightly flexuous, slightly narrower and truncate or rounded at the apex, 6.8–13 µm long pedicellate, with or without a barrel- or jar-shaped apical ring. *Ascospores* 13–17 × 4.2–5.6 µm ($\bar{x}$ = 15 × 4.8 µm, n = 30), overlapping, oblique uniseriate, fusoid, rounded at both ends, straight, with three inconspicuous septa, slightly constricted at the middle septum, guttulate, hyaline, thin and smooth-walled, without a gelatinous sheath.

Culture characteristics: Ascospores germinate on the PDA within 24 h, with germ tubes produced from the end. Conidia germinate on the PDA within 24 h, with germ tubes produced from both ends. Colonies growing on the PDA after 4 weeks of incubation at room temperature attain a diameter of about 15 mm. Mycelia dry and dense. Colonies ate semi-immersed in PDA, with irregular edges, rough surfaces, protrusions at the center, and celadon. The reverse is dark green and smooth.

Material examined: China, Yunnan Province, Qujing City, Luoping County (25°01′52.57″ N; 104°42′47.40″ E), on submerged decaying wood, 15 July 2023, Wen-Peng Wang, S-5446 (HKAS 131980, holotype), ex-type culture CGMCC 3.27019 = KUNCC 23–15650; Wenshan Zhuang and Miao Autonomous Prefecture, Guangnan County (24°11′53.10″ N; 104°92′82.31″ E), on submerged decaying wood, 19 July 2023, Wen-Peng Wang, S-5540 (HKAS 131972, paratype), living culture KUNCC 23–16569.

Notes: *Pleurotheciella obliqua* is the fourth sexual species in *Pleurotheciella*, the other three species are *P. erumpens*, *P. fusiformis*, and *P. rivularia* [4,12,15]. *P. obliqua* is distinctly different from *P. erumpens* and *P. fusiformis* with fusoid and shorter ascospores, but resembles *P. rivularia* with cylindrical-to-clavate, truncate apical, short stipitate asci with an apical ring, and smaller ascospores without a gelatinous sheath. However, *Pleurotheciella obliqua* has subcylindrical asci, and obliquely uniseriate, straight, inconspicuously septate ascospores, which are different from *P. rivularia*. In the phylogenetic analysis, *Pleurotheciella obliqua* (CGMCC 3.27019 and KUNCC 23–16569) is sister to *P. longidenticulata* (CGMCC 3.27018) and *P. saprophytica* (MFLUCC 16–1251) with 93% ML/1.00 PP support (Figure 1). *P. obliqua* differs from *P. longidenticulata* without irregular expansion and constricted conidiophores, the denticles shorter (2.2 vs. 3.7 µm) and the conidia are clavate-to-subcylindrical. *P. obliqua* differs from *P. saprophytica* by longer conidiophores (32–100 vs. 44–52 µm) and obtuse basal conidia [12].

***Rhexoacrodictys melanospora*** S.X. Bao, R.J. Xu & Q. Zhao, Phytotaxa 594 (3): 217 (2023), Figure 10.

Fungal Names number: FN 559983.

*Saprobic* on submerged decaying wood. **Asexual morph**: *Colonies* superficial, effuse, hairy, solitary or in small groups, black, with smooth conidia at the apex of conidiophores. *Mycelium* mostly immersed, composed of branched, septate, smooth-walled, pale brown hyphae. *Conidiophores* 41–87 × 4.5–7.1 µm ($\bar{x}$ = 61 × 5.8 µm, n = 20), macronematous, mononematous, erect, straight, 3–5-septate, cylindrical, slightly tapering towards the tip, thick, smooth-walled, brown, with darkened septa. *Conidiogenous cells* 3.8–6.2 (–8) × 2.9–4.5 µm ($\bar{x}$ = 5.2 × 3.8 µm, n = 20), monoblastic, integrated, terminal, pale brown to brown, cylindrical, sometimes with subhyaline percurrent extensions. *Conidia* 27–39 × 14–20 µm ($\bar{x}$ = 31 × 16.8 µm, n = 20), solitary, acrogenous, obovoid, muriform, transversely and longitudinally septate, with transverse septa typically spanning the whole conidial width, and incomplete longitudinal septa, subhyaline to pale brown when immature, becoming dark brown to black at maturity, thin, smooth-walled, and easily breaking in water, with a 2–4.9 µm long cuneiform base. **Sexual morph**: Undetermined.

Culture characteristics: Conidia germinate on PDA within 24 h, with germ tubes produced from the base. Colonies grow on PDA after 3 weeks of incubation at room temperature, attaining a diameter of about 15 mm. Mycelia dry and dense. Colonies on the surface of PDA protrude, with regular paler edges, and are gray-green. The reverse is dark green and smooth. *Conidiophores* reduced to conidiogenous cells. *Conidiogenous cells* 6.5–26 × 2.3–5.5 µm ($\bar{x}$ = 13.9 × 3.5 µm, n = 20), monoblastic, integrated, hyaline to pale brown, cylindrical, smooth. *Conidia* 19–33 × 12–23 µm ($\bar{x}$ = 24 × 16.7 µm, n = 30), broad oval to subglobose, muriform, constricted at all the septum, hyaline when young, olive green to black when mature, irregular transversely and longitudinally septate.

Material examined: China, Yunnan Province, Honghe Hani and Yi Autonomous Prefecture, Mile City (24°42′69.75″ N; 103°48′34.68″ E), on submerged decaying wood, 14 July 2023, Wen-Peng Wang S-5432 (HKAS 131977), living culture KUNCC 23–16529.

Notes: Phylogenetic analysis showed that our new collection (KUNCC 23–16529) clustered with the clade of *Rhexoacrodictys melanospora* (Figure 1). Our new collection morphologically resembles *R. melanospora* in having erect, straight, or slightly flexuous conidiophores with percurrently extending conidiogenous cells, and obovoid, muriform, transversely and longitudinally septate conidia [24,52]. We, therefore, identified our new collection as *Rhexoacrodictys melanospora*, based on morphological and phylogenetic analysis, which was the first time to reported from a freshwater habitat.

## 4. Discussion

*Phaeoisaria* was established by Höhnel [11]: thirty-six epithets are listed in Index Fungorum [20], and only one species is known as a sexual morph. The asexual morph of *Phaeoisaria* is characterized by compactly adpressed synnematous conidiophores, which are subcylindrical, curved conidiogenous cells with multiple denticulate conidiogenous loci, and ellipsoidal-to-obovoid or clavate, aseptate, hyaline conidia; the asexual morph exhibited little morphological differences between the species in the genus [13,16,50]. Therefore, species of this genus are distinguished primarily by molecular evidence and supplemented by the sizes of synnemata, conidiogenous cells, and conidia [9,13,53].

In this study, we observed the reproduced asexual morph of *Phaeoisaria synnematica* on the PDA medium, and we also observed produced chlamydospores of this species but did not find the reduced single conidiogenous cell arising from aerial hyphae conidiophores and septate conidia, which were mentioned by Boonmee et al. [16]. In addition, five species, including *P. annesophieae*, *P. fasciculata*, *P. goiasensis*, *P. guttulate*, and *P. loranthacearum,* also observed some morphologies in the culture that differed from other species of this genus, including producing chlamydospores, a lack of synnemata, either reduced or short, arising from aerial hyphae conidiophores, and septate conidia [1,9,17,54,55]. However, these characteristics were not observed in all species on the surface of the natural substrates [12,16,49,56,57]. Therefore, the difference in growth substrates has a great influence on the morphogenesis of species in this genus, and we should pay more attention to the colony on culture to provide as many morphological characteristics of the same species as possible and provide more evidence to identify the species of *Phaeoisaria*.

We compiled fifty-seven freshwater *Pleurotheciaceae* species belonging to eleven genera, as shown in Table 2, and these species were collected from different climatic zones around the world. These results indicate that the species of this family have strong adaptability to different climates, and freshwater habitats are their main living environment (Figure 1). Asia is the region with the highest number of reported species, especially in Yunnan Province, China, where more than half (29/57, 50.88%) of these species are found in Yunnan Province. The suitable, complex, and changeable climate and geographical environment of Yunnan Province provide corresponding growth conditions for different variations in organisms. This is true not only for species of *Pleurotheciaceae* but also for other fungal taxa [29,30,58].

Yunnan Province is the region with the highest concentration of *Pleurotheciaceae* species [12,14,59,60], but most *Pleurotheciaceae* species are collected from the western and southern parts of Yunnan Province [14,28,29,58]. This is a study of freshwater *Pleurotheciaceae* that was carried out specifically in karst landforms of Yunnan Province; in this study, nine freshwater *Pleurotheciaceae* species are reported for the first time in Yunnan Province, and these species were all collected from eastern Yunnan Province. Therefore, the eastern part of Yunnan Province is also extremely rich in freshwater fungal resources compared with other parts of Yunnan Province, but there is a lack of systematic research.

## Figures and Tables

**Figure 1 jof-10-00516-f001:**
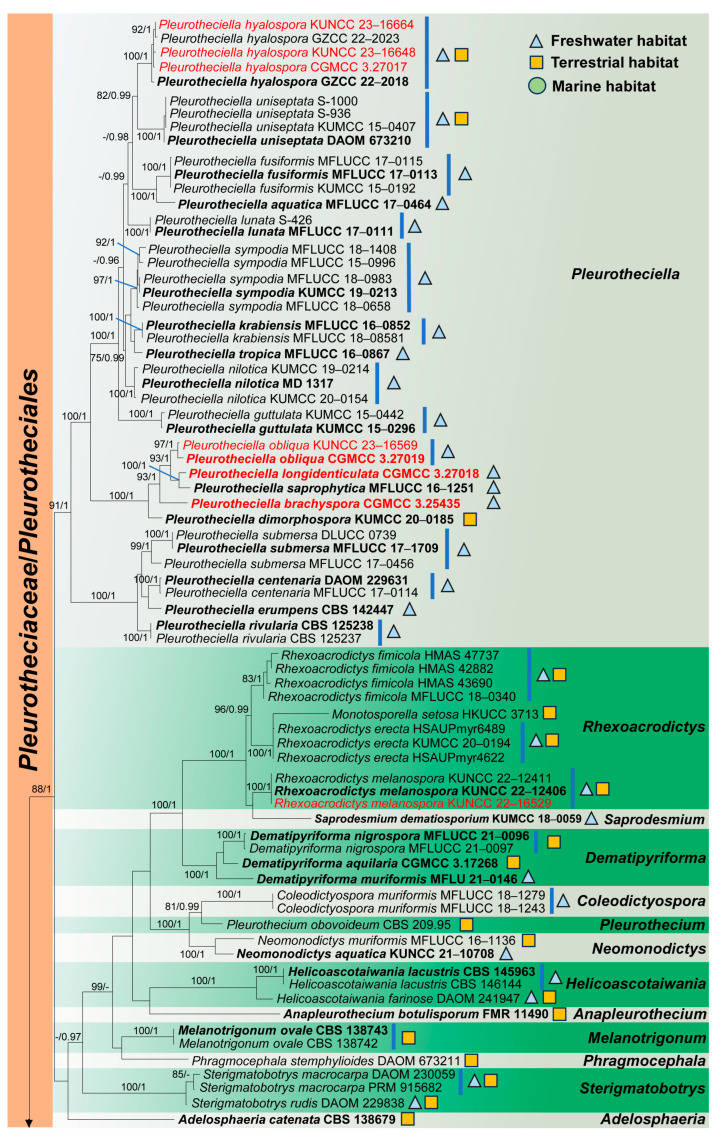
RAxML tree based on combined LSU, ITS, SSU, and *rpb*2 sequence data of *Pleurotheciaceae* (*Pleurotheciales*) and *Savoryellaceae* (*Savoryellales*). Bootstrap support values for maximum likelihood (ML) greater than 75% and Bayesian posterior probabilities (PP) greater than 0.95 are given as ML/PP above the nodes. Newly obtained sequences are indicated in red, and ex-type strains are in bold.

**Figure 2 jof-10-00516-f002:**
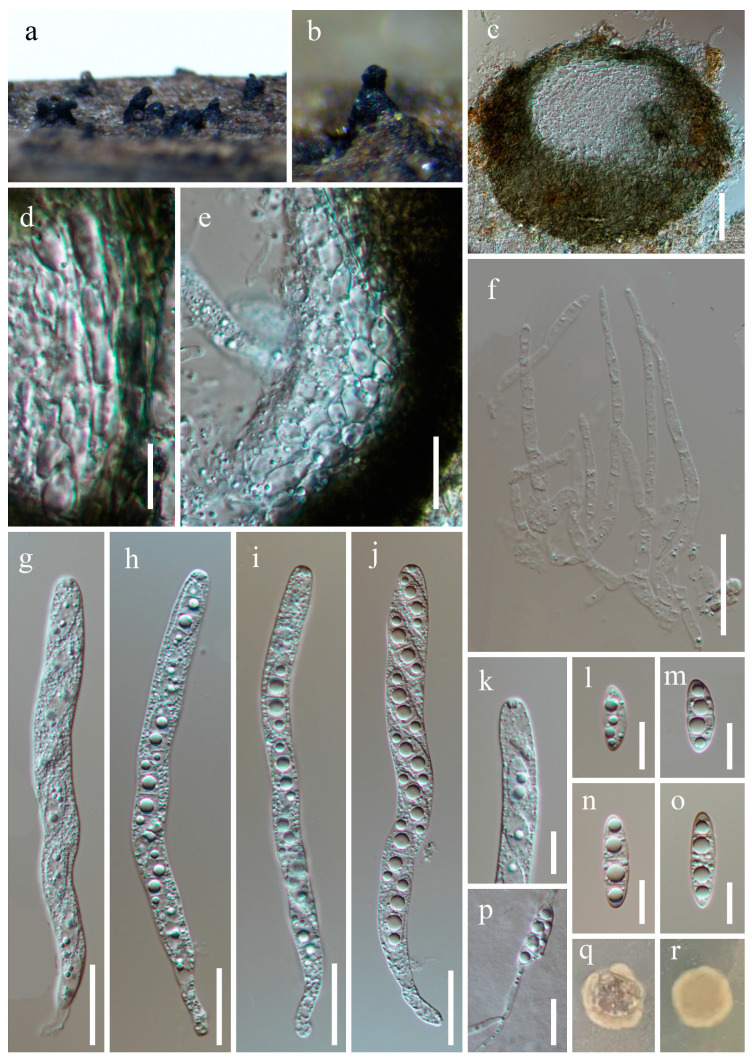
*Obliquifusoideum triseptatum* (HKAS 131970, holotype). (**a**,**b**)—Ascomata on the substratum. (**c**)—Vertical section of ascoma. (**d**,**e**)—Structure of peridium. (**f**)—Paraphyses. (**g**–**j**)—Asci. (**k**)—Apex of ascus. (**l**–**o**)—Ascospores. (**p**)—Germinating ascospore. (**q**,**r**)—Colony on PDA from the surface and reverse. Scale bars: (**c**) = 40 µm, (**d**,**k**–**p**) = 10 µm, and (**e**–**j**) = 20 µm.

**Figure 3 jof-10-00516-f003:**
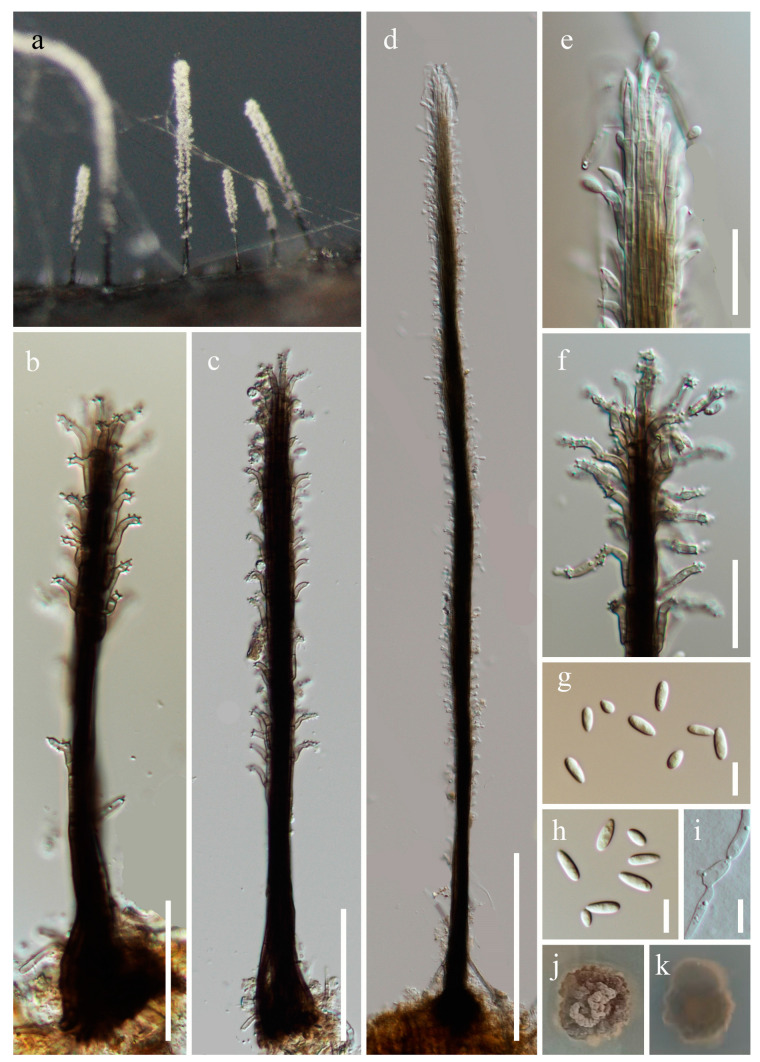
*Phaeoisaria obovata* (HKAS 131983, holotype). (**a**)—Colonies on the substratum. (**b**–**d**)—Synnemata. (**e**,**f**)—Apex of synnemata. (**g**,**h**)—Conidia. (**i**)—Germinating conidium. (**j**,**k**)—Colony on PDA from surface and reverse. Scale bars: (**b**) = 30 µm, (**c**) = 50 µm, (**d**) = 150 µm, (**e**,**f**) = 20 µm, and (**g**–**i**) = 7 µm.

**Figure 4 jof-10-00516-f004:**
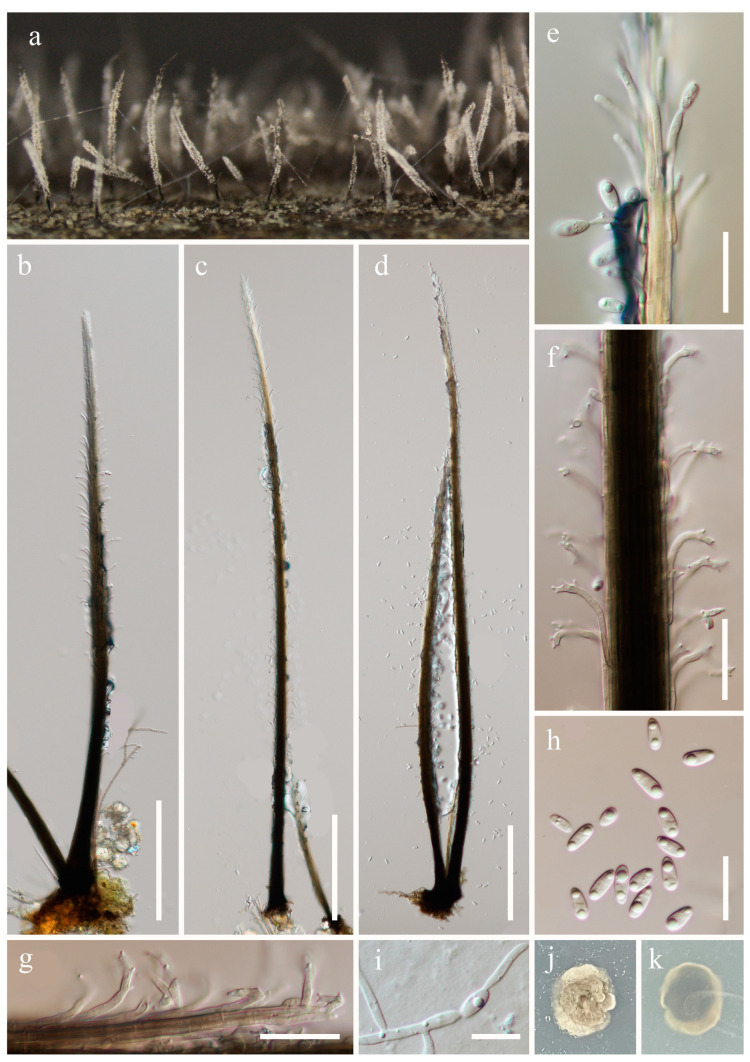
*Phaeoisaria sedimenticola* (HKAS 131978). (**a**)—Colonies on the substratum. (**b**–**d**)—Synnemata. (**e**)—Apex of synnema with conidia. (**f**,**g**)—Conidiogenous cells. (**h**)—Conidia. (**i**)—Germinating conidium. (**j**,**k**)—Colony on PDA from the surface and reverse. Scale bars: (**b**–**d**) = 150 µm, (**e**–**g**) = 20 µm, (**h**) = 15 µm, and (**i**) = 10 µm.

**Figure 5 jof-10-00516-f005:**
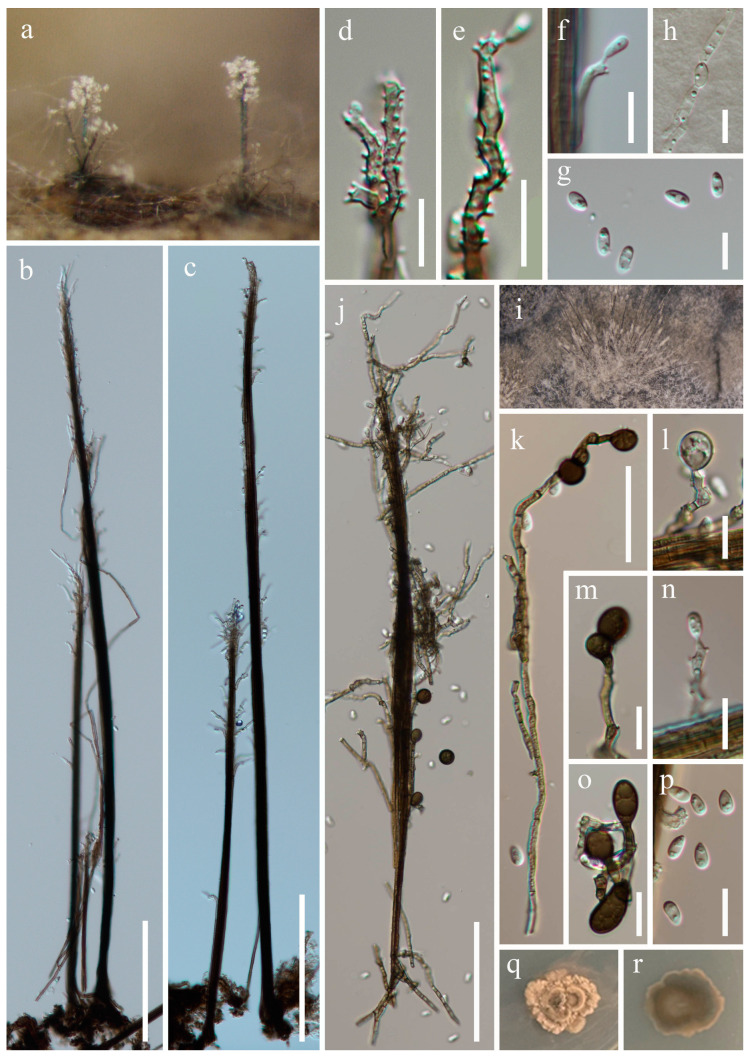
*Phaeoisaria synnematica.* (**a**–**c**,**f**–**r**) HKAS 131965; (**d**,**e**) HKAS 131974. (**a**)—Colonies on the substratum. (**b**,**c**)—Synnemata. (**d**–**f**)—Conidiogenous cells. (**g**)—Conidia. (**h**)—Germinating conidium. (**i**–**p**)—Re-produced asexual morph. (**i**)—Colonies on PDA. (**j**)—Synnema. (**k**)—Conidiophores with chlamydospores. (**l**,**m**,**o**)—Chlamydospores. (**n**)—Conidiogenous cell with conidia. (**p**)—conidia. (**q**,**r**)—Colony on PDA from the surface and reverse. Scale bars: (**b**,**c**) = 100 µm, (**d**,**e**,**g**,**l**–**p**) = 10 µm, (**f**,**h**) = 7 µm, (**j**) = 70 µm, and (**k**) = 30 µm.

**Figure 6 jof-10-00516-f006:**
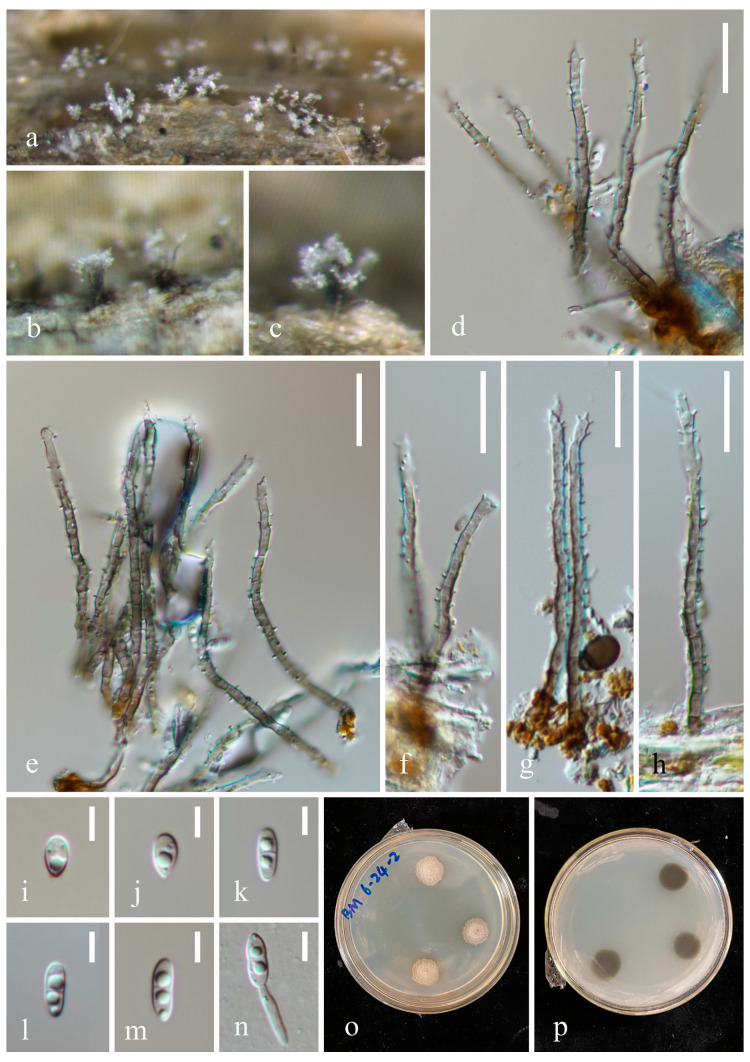
*Pleurotheciella brachyspora* (HKAS 131981, holotype). (**a**–**c**)—Colonies on the substratum. (**d**–**h**)—Conidiophores. (**i**–**m**)—Conidia. (**n**)—Germinating conidium. (**o**,**p**)—Colonies on PDA from the surface and reverse. Scale bars: (**d**–**h**) = 20 µm, and (**i**–**n**) = 5 µm.

**Figure 7 jof-10-00516-f007:**
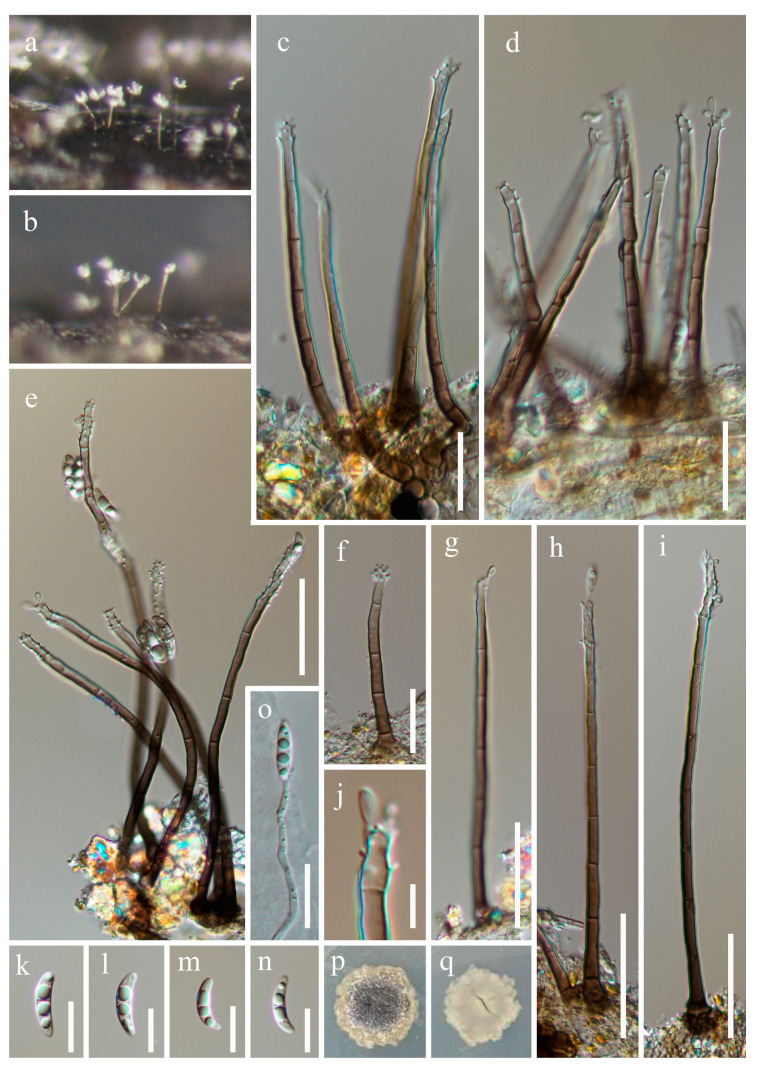
*Pleurotheciella hyalospora.* (**a**,**c**,**d**) HKAS 131975, (**b**,**f**,**h**) HKAS 131982, (**e**,**g**,**i**–**q**) HKAS 131979). (**a**,**b**)—Colonies on the substratum. (**c**–**e**)—Gathered in small groups of conidiophores. (**f**–**i**)—Solitary of conidiophores. (**j**)—Apex of conidiophore with conidiogenous cells. (**k**–**n**)—Conidia. (**o**)—Germinating conidium. (**p**,**q**)—Colony on PDA from the surface and reverse. Scale bars: (**c**,**d**) = 20 µm, (**e**,**g**–**i**) = 30 µm, (**f**,**o**) = 15 µm, (**j**) = 5 µm, and (**k**–**n**) = 10 µm.

**Figure 8 jof-10-00516-f008:**
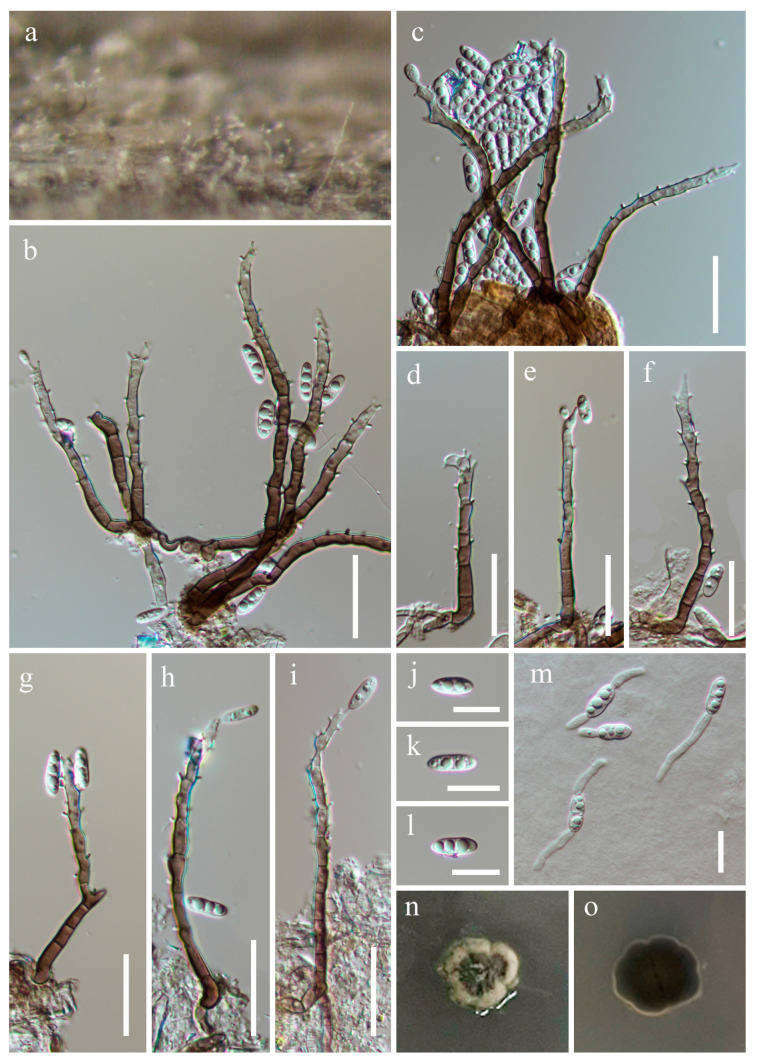
*Pleurotheciella longidenticulata.* (HKAS 131973, holotype) (**a**)—Colonies on the substratum. (**b**–**i**)—Conidiophores, conidiophores with conidia. (**j**–**l**)—Conidia. (**m**)—Germinating conidia. (**n**,**o**)—Colony on PDA from the surface and reverse. Scale bars: (**c**–**i**) = 20 µm, and (**j**–**m**) = 10 µm.

**Figure 9 jof-10-00516-f009:**
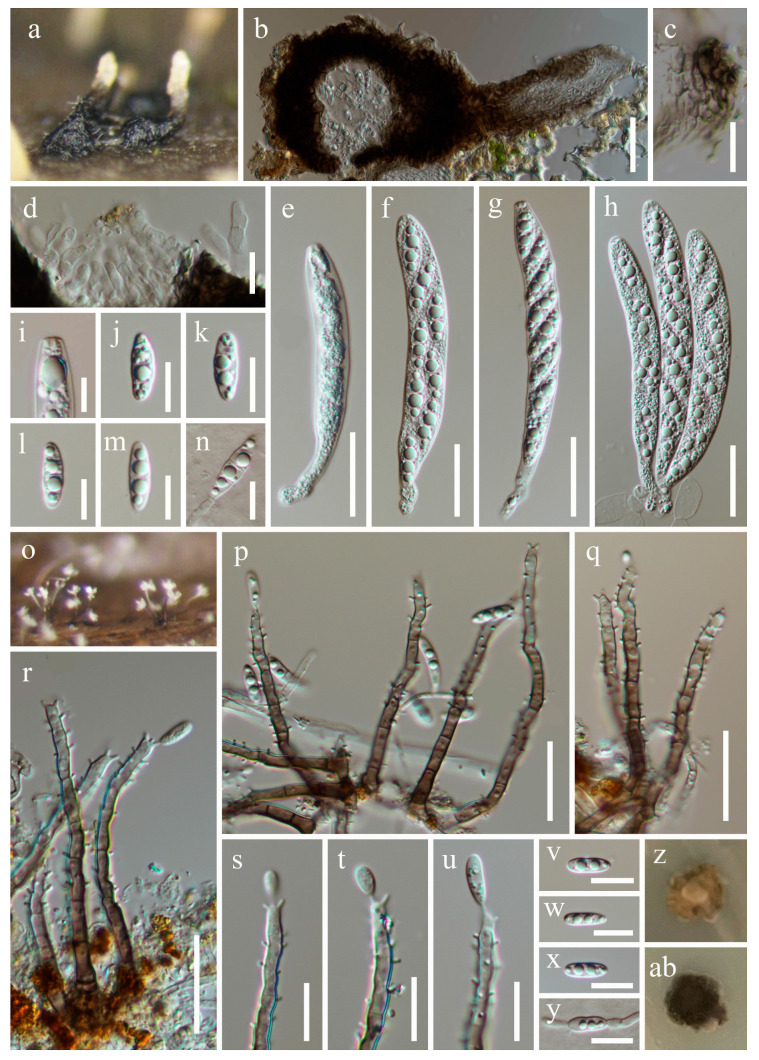
*Pleurotheciella obliqua.* (**a**–**n**) HKAS 131980, holotype, (**o**–**ab**) HKAS 131972) (**a**)—Ascomata on the substratum. (**b**)—Vertical section of ascoma. (**c**)—Structure of peridium. (**d**)—Paraphyses. (**e**–**h**)—Asci. (**i**)—Apex of ascus. (**j**–**m**)—Ascospores. (**n**)—Germinating ascospore. (**o**)—Colonies on the substratum. (**p**–**r**)—Conidiophores. (**s**–**u**)—Conidiogenous cells with conidia. (**v**–**x**)—Conidia. (**y**)—Germinating conidium. (**z**,**ab**)—Colony on PDA from surface and reverse. Scale bars: (**b**) = 25 µm, (**c**,**d**,**j**–**n**,**s**–**y**) = 10 µm, (**e**–**h**,**p**–**r**) = 20 µm, and (**i**) = 5 µm.

**Figure 10 jof-10-00516-f010:**
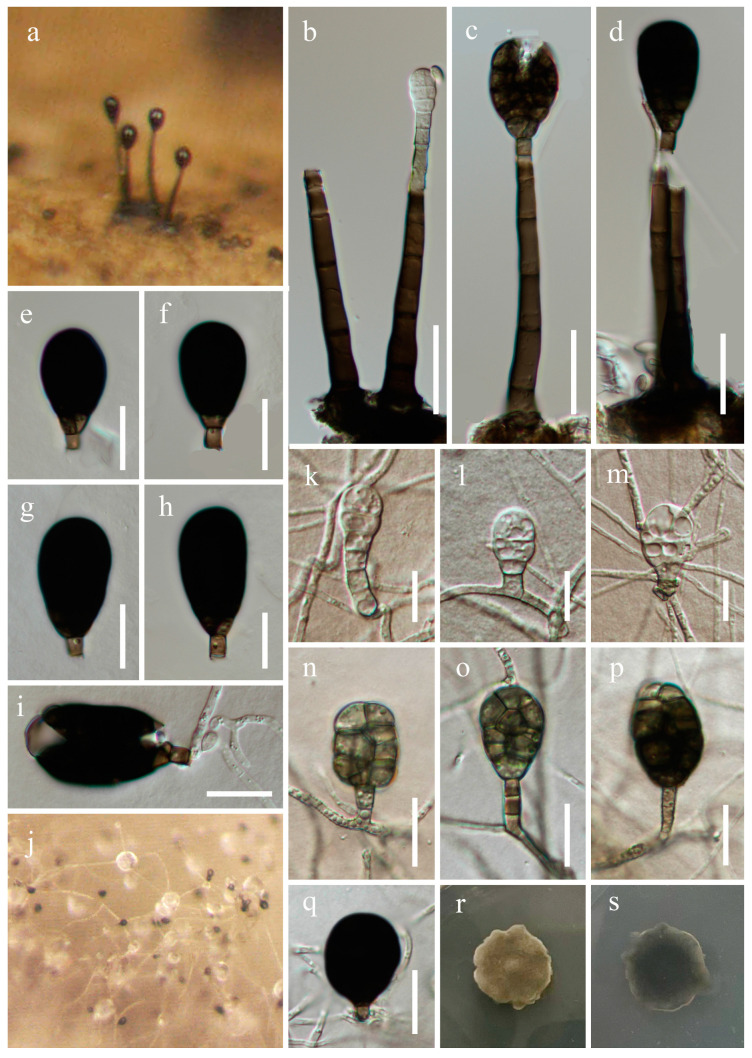
*Rhexoacrodictys melanospora* (HKAS 131977). (**a**)—Colonies on the substratum. (**b**–**d**)—Conidiophores with conidia. (**e**–**h**)—Cnidia. (**i**)—Germinating conidium. (**j**–**q**)—Re-produced on PDA. (**j**)—Colony on PDA. (**k**–**q**)—Conidiogenous cells with conidia. (**r**,**s**)—Colony on PDA from the surface and reverse. Scale bars: (**b**–**d**) = 20 µm, (**e**–**i**,**n**–**q**) = 15 µm, and (**k**–**m**) = 10 µm.

**Table 1 jof-10-00516-t001:** Strains/specimens used for phylogenetic analysis and their GenBank accession numbers. The ex-type species, strains, and sequences are in bold, with those newly generated indicated in red.

Species	Strains/Specimens	GenBank Accession Number			
		LSU	ITS	SSU	*rpb*2
** *Adelosphaeria catenata* **	**CBS 138679**	**KT278707**	**KT278721**	**KT278692**	**KT278743**
** *Anapleurothecium botulisporum* **	**FMR 11490**	**KY853483**	**KY853423**	**_**	**_**
*Canalisporium exiguum*	SS 00809	GQ390281	GQ390296	GQ390266	HQ446436
** *Canalisporium grenadoideum* **	**SS 03615**	**GQ390267**	**_**	**GQ390252**	**HQ446420**
*Canalisporium pulchrum*	SS 03982	GQ390277	GQ390292	GQ390262	HQ446432
*Coleodictyospora muriformis*	MFLUCC 18–1243	MW981648	MW981642	MW981704	_
*Coleodictyospora muriformis*	MFLUCC 18–1279	MW981649	MW981643	MW981705	_
** *Conioscypha lignicola* **	**CBS 335.93**	**AY484513**	**_**	**JQ437439**	**JQ429260**
** *Conioscypha minutispora* **	**CBS 137253**	**MH878131**	**_**	**_**	**_**
** *Dematipyriforma aquilaria* **	**CGMCC 3.17268**	**KJ138623**	**KJ138621**	**KJ138622**	**_**
** *Dematipyriforma muriformis* **	**MFLU 21–0146**	**OM654770**	**OM654773**	**_**	**_**
** *Dematipyriforma nigrospora* **	**MFLUCC 21–0096**	**MZ538558**	**MZ538524**	**_**	**_**
*Dematipyriforma nigrospora*	MFLUCC 21–0097	MZ538559	MZ538525	MZ538574	MZ567113
*Helicoascotaiwania farinosa*	DAOM 241947	JQ429230	JQ429145	_	_
** *Helicoascotaiwania lacustris* **	**CBS 145963**	**MN699430**	**MN699399**	**MN699382**	**MN704304**
*Helicoascotaiwania lacustris*	CBS 146144	MN699432	MN699401	MN699384	MN704306
*Melanotrigonum ovale*	CBS 138743	KT278709	KT278724	KT278696	KT278745
** *Melanotrigonum ovale* **	**CBS 138742**	**KT278708**	**KT278723**	**KT278695**	**KT278744**
*Monotosporella setosa*	HKUCC 3713	AF132334	_	_	_
** *Nenascotaiwania fusiformis* **	**MFLU 15–1156**	**NG_057114**	**MG388215**	**_**	**_**
*Neoascotaiwania fusiformis*	MFLUCC 15–0625	KX550894	_	KX550898	_
** *Neomondictys aquatica* **	**KUNCC 21–10708**	**OK245417**	**MZ686200**	**_**	**_**
*Neomonodictys muriformis*	MFLUCC 16–1136	MN644485	MN644509	_	_
** *Obliquifusoideum guttulatum* **	**MFLUCC 18–1233**	**MW981650**	**MW981645**	**MW981706**	**_**
* **Obliquifusoideum triseptatum** *	**CGMCC 3.27014**	**PP049503**	**PP445243**	**PP049521**	**PP068779**
** *Phaeoisaria annesophieae* **	**CBS 143235**	**MG022159**	**MG022180**	**_**	**_**
*Phaeoisaria annesophieae*	MFLU 19–0531	MT559084	MT559109	_	_
** *Phaeoisaria aquatica* **	**MFLUCC 16–1298**	**MF399254**	**MF399237**	**_**	**MF401406**
*Phaeoisaria clematidis*	MFLUCC 16–1273	MF399246	MF399229	_	_
*Phaeoisaria clematidis*	MFLUCC 17–1341	MF399247	MF399230	MF399216	MF401400
*Phaeoisaria clematidis*	MFLUCC 17–1968	MG837017	MG837022	MG837027	_
*Phaeoisaria clematidis*	DAOM 226789	JQ429231	JQ429155	JQ429243	JQ429262
** *Phaeoisaria dalbergiae* **	**CPC 39540**	**OK663742**	**OK664703**	**OK663796**	**OK651159**
** *Phaeoisaria ellipsoidea* **	**IFRDCC 3134**	**ON533387**	**ON533383**	**_**	**_**
** *Phaeoisaria fasciculata* **	**CBS 127885**	**KT278705**	**KT278719**	**KT278693**	**KT278741**
*Phaeoisaria fasciculata*	DAOM 230055	KT278706	KT278720	KT278694	KT278742
** *Phaeoisaria filiformis* **	**MFLUCC 18–0214**	**MK835852**	**MK878381**	**MK834785**	**_**
*Phaeoisaria filiformis*	KUNCC 23–13723	OR600967	OR589319	OR743212	OR820908
** *Phaeoisaria goiasensis* **	**FCCUFG 02**	**MT375865**	**MT210320**	**_**	**_**
*Phaeoisaria goiasensis*	FCCUFG 03	MT375866	MT210321	_	_
** *Phaeoisaria guttulata* **	**MFLUCC 17–1965**	**MG837016**	**MG837021**	**MG837026**	**_**
** *Phaeoisaria laianensis* **	**CCTCC AF2022069**	**ON937557**	**ON937559**	**ON937562**	**_**
*Phaeoisaria laianensis*	CCTCC AF2022073	ON937561	ON937560	ON937558	_
** *Phaeoisaria loranthacearum* **	**CBS 140009**	**MH878676**	**KR611888**	**_**	**_**
*Phaeoisaria loranthacearum*	BYCDW25	_	MG820097	_	_
*Phaeoisaria loranthacearum*	BYCDW24	_	MG820098	_	_
** *Phaeoisaria motuoensis* **	**KUNCC 10410**	**OQ947034**	**OP626333**	**OQ947036**	**_**
*Phaeoisaria motuoensis*	KUNCC 10450	OQ947035	OQ947032	OQ947037	_
*Phaeoisaria motuoensis*	KUNCC 23–15489	PP049517	PP049499	PP049535	PP068781
*Phaeoisaria motuoensis*	KUNCC 23–15451	PP049518	PP049500	PP049536	PP068782
** *Phaeoisaria microspora* **	**MFLUCC 16–0033**	**MF167351**	**MF671987**	**_**	**MF167352**
*Phaeoisaria microspora*	KUNCC 23–15498	PP049519	PP049501	PP049537	_
* **Phaeoisaria obovata** *	**CGMCC 3.27015**	**PP049504**	**PP049488**	**PP049522**	**PP068788**
* Phaeoisaria obovata *	KUNCC 23–15598	PP049505	PP049489	PP049523	PP068784
** *Phaeoisaria pseudoclematidis* **	**MFLUCC 11–0393**	**KP744501**	**KP744457**	**KP753962**	**_**
*Phaeoisaria pseudoclematidis*	KUNCC 23–13718	OR600968	OR589320	_	_
** *Phaeoisaria sedimenticola* **	**CGMCC3.14949**	**JQ031561**	**JQ074237**	**_**	**_**
*Phaeoisaria sedimenticola*	S-908	MK835851	MK878380	_	_
* Phaeoisaria sedimenticola *	KUNCC 23–14648	PP049506	PP049490	PP049524	PP068783
* Phaeoisaria sedimenticola *	KUNCC 23–15613	PP049507	PP049491	PP049525	PP068785
** *Phaeoisaria siamensis* **	**MFLUCC 16–0607**	**MK607613**	**MK607610**	**MK607612**	**MK607611**
*Phaeoisaria sparsa*	FMR 11939	HF677185	_	_	_
** *Phaeoisaria synnematica* **	**NFCCI 4479**	**MK391492**	**MK391494**	**_**	**_**
* Phaeoisaria synnematica *	KUNCC 23–16573	PP049508	PP049492	PP049526	PP068786
* Phaeoisaria synnematica *	KUNCC 23–16619	PP049509	PP049493	PP049527	PP068787
*Phragmocephala stemphylioides*	KAS 4277	KT278717	KT278730	_	_
* **Pleurotheciella aquatica** *	**MFLUCC 17–0464**	**MF399253**	**MF399236**	**MF399220**	**MF401405**
* **Pleurotheciella brachyspora** *	**CGMCC 3.25435**	**OR600969**	**OR589321**	**PP049532**	**PP068773**
** *Pleurotheciella centenaria* **	**DAOM 229631**	**JQ429234**	**JQ429151**	**JQ429246**	**JQ429265**
*Pleurotheciella centenaria*	MFLUCC 17–0114	MK835849	_	_	MN194027
** *Pleurotheciella dimorphospora* **	**KUMCC 20–0185**	**MW981444**	**MW981446**	**MW981454**	**MZ509665**
** *Pleurotheciella erumpens* **	**CBS 142447**	**MN699435**	**MN699406**	**MN699387**	**MN704311**
** *Pleurotheciella fusiformis* **	**MFLUCC 17–0113**	**MF399250**	**MF399233**	**MF399218**	**MF401403**
*Pleurotheciella fusiformis*	MFLUCC 17–0115	MF399249	MF399232	MF399217	MF401402
*Pleurotheciella fusiformis*	KUMCC 15–0192	MF399251	MF399234	_	_
*Pleurotheciella guttulata*	KUMCC 15–0442	MF399256	MF399239	MF399222	MF401408
** *Pleurotheciella guttulata* **	**KUMCC 15–0296**	**MF399257**	**MF399240**	**MF399223**	**MF401409**
** *Pleurotheciella hyalospora* **	**GZCC 22–2018**	**OQ002371**	**OQ002374**	**OQ002377**	**OP999221**
*Pleurotheciella hyalospora*	GZCC 22–2023	OQ002370	OQ002373	OQ002376	OP999220
* Pleurotheciella hyalospora *	CGMCC 3.27017	PP049510	PP049494	PP049528	PP068775
* Pleurotheciella hyalospora *	KUNCC 23–16648	PP049511	PP049495	PP049529	PP068778
* Pleurotheciella hyalospora *	KUNCC 23–16664	PP049512	PP445244	PP049530	PP068780
** *Pleurotheciella krabiensis* **	**MFLUCC 16–0852**	**MG837013**	**MG837018**	**MG837023**	**_**
*Pleurotheciella krabiensis*	MFLUCC 18–0858	MG837014	MG837019	MG837024	_
* **Pleurotheciella longidenticulata** *	**CGMCC 3.27018**	**PP049513**	**PP049496**	**PP049531**	**PP068776**
** *Pleurotheciella lunata* **	**MFLUCC 17–0111**	**MF399255**	**MF399238**	**MF399221**	**MF401407**
*Pleurotheciella lunata*	S-426	MK835847	MK878378	MK834782	_
** *Pleurotheciella nilotica* **	**MD 1317**	**KX611344**	**_**	**MN356449**	**_**
*Pleurotheciella nilotica*	KUMCC 19–0214	MT559087	MT555416	MT559095	_
*Pleurotheciella nilotica*	KUMCC 20–0154	MT559121	MT555417	MT555733	_
* **Pleurotheciella obliqua** *	**CGMCC 3.27019**	**PP049514**	**PP049497**	**PP049533**	**PP068774**
* Pleurotheciella obliqua *	KUNCC 23–16569	PP049515	PP049498	PP049534	PP068777
*Pleurotheciella rivularia*	CBS 125237	JQ429233	JQ429161	JQ429245	JQ429264
** *Pleurotheciella rivularia* **	**CBS 125238**	**JQ429232**	**JQ429160**	**JQ429244**	**JQ429263**
** *Pleurotheciella saprophytica* **	**MFLUCC 16–1251**	**MF399258**	**MF399241**	**MF399224**	**MF401410**
** *Pleurotheciella submersa* **	**MFLUCC 17–1709**	**MF399260**	**MF399243**	**MF399226**	**MF401412**
*Pleurotheciella submersa*	MFLUCC 17–0456	MF399261	MF399244	MF399227	MF401413
*Pleurotheciella submersa*	DLUCC 0739	MF399259	MF399242	MF399225	MF401411
*Pleurotheciella sympodia*	MFLUCC 18–1408	MW981652	MW981644	_	_
*Pleurotheciella sympodia*	MFLUCC 15–0996	MW981651	MW981641	MW981703	_
*Pleurotheciella sympodia*	MFLUCC 18–0658	MT559086	MT555418	MT559094	_
*Pleurotheciella sympodia*	MFLUCC 18–0983	MT555425	MT555419	MT555734	_
** *Pleurotheciella sympodia* **	**KUMCC 19–0213**	**MT555426**	**MT555420**	**_**	**_**
** *Pleurotheciella tropica* **	**MFLUCC 16–0867**	**MG837015**	**MG837020**	**MG837025**	**_**
*Pleurotheciella uniseptata*	S-1000	MK835845	MK878376	_	MN194024
*Pleurotheciella uniseptata*	S-936	MK835846	MK878377	MK834781	MN194025
*Pleurotheciella uniseptata*	KUMCC 15–0407	MF399248	MF399231	_	MF401401
** *Pleurotheciella uniseptata* **	**DAOM 673210 **	**KT278716**	**KT278729**	**_**	**_**
** *Pleurothecium aquaticum* **	**MFLUCC 17–1331**	**MF399263**	**MF399245**	**_**	**_**
*Pleurothecium aquaticum*	MFLUCC 21–0148	OM654772	OM654775	OM654807	_
*Pleurothecium obovoideum*	CBS 209.95	EU041841	EU041784	_	_
** *Pleurothecium pulneyense* **	**MFLUCC 16–1293**	**MF399262**	**_**	**MF399228**	**MF401414**
*Pleurothecium semifecundum*	CBS 131482	JQ429239	JQ429158	JQ429253	_
** *Pleurothecium semifecundum* **	**CBS 131271**	**JQ429240**	**JQ429159**	**JQ429254**	**JQ429270**
*Rhexoacrodictys erecta*	HSAUPmyr4622	KX033556	KU999964	KX033526	_
*Rhexoacrodictys erecta*	HSAUPmyr6489	KX033555	KU999963	KX033525	_
*Rhexoacrodictys erecta*	KUMCC 20–0194	MT559123	MT555421	_	_
*Rhexoacrodictys fimicola*	HMAS42882	KX033554	KU999962	KX033524	_
*Rhexoacrodictys fimicola*	HMAS43690	KX033550	KU999957	KX033519	_
*Rhexoacrodictys fimicola*	HMAS47737	KX033553	KU999960	KX033522	_
*Rhexoacrodictys fimicola*	MFLUCC 18–0340	OM654771	OM654774	OM654806	_
** *Rhexoacrodictys melanospora* **	**KUNCC 22–12406**	**OP168087**	**OP168085**	**OP168088**	**OP208807**
*Rhexoacrodictys melanospora*	KUNCC 22–12411	OP168101	OP168093	OP168099	OP208808
* Rhexoacrodictys melanospora *	KUNCC 23–16529	PP049516	_	_	_
** *Saprodesmium dematiosporium* **	**KUMCC 18–0059**	**MW981647**	**MW981646**	**MW981707**	**_**
*Sterigmatobotrys macrocarpa*	DAOM 230059	GU017316	JQ429154	_	_
*Sterigmatobotrys macrocarpa*	PRM 915682	GU017317	JQ429153	JQ429255	_
*Sterigmatobotrys rudis*	DAOM 229838	JQ429241	JQ429152	JQ429256	JQ429272

**Table 2 jof-10-00516-t002:** Species of *Pleurotheciaceae* from freshwater habitats.

Species	Host	Country	Reference
*Coleodictyospora muriformis*	unknown submerged wood	Thailand	[6]
*Dematipyriforma aquatica*	unknown submerged wood	Egypt	[9]
*Dematipyriforma globispora*	unknown submerged wood	Egypt	[9]
*Dematipyriforma muriformis*	unknown submerged wood	Thailand	[8]
*Dematipyriforma nilotica*	submerged date palm rachis	Egypt	[9]
*Helicoascotaiwania farinosa*	unknown submerged wood	USA	[4]
*Helicoascotaiwania lacustris*	submerged branch of *Populus* sp. and *Salix atrocinerea*	France	[4]
*Neomonodictys aquatica*	unknown submerged wood	China (Yunnan Province)	[59]
*Obliquifusoideum guttulatum*	unknown submerged wood	Thailand	[6]
*Obliquifusoideum triseptatum*	unknown submerged wood	China (Yunnan Province)	This study
*Phaeoisaria aguilerae*	unknown submerged twig	Cuba	[30]
*Phaeoisaria annesophieae*	unknown submerged wood	Thailand	[7]
*Phaeoisaria aquatica*	unknown submerged wood	China (Yunnan Province)	[12]
*Phaeoisaria clematidis*	submerged wood of various species	China (Yunnan Province, Xizang), ThailandPhilippines, Brazil	[12,30,51]
*Phaeoisaria ellipsoidea*	unknown submerged wood	China (Yunnan Province)	[60]
*Phaeoisaria guttulata*	unknown submerged wood	China (Guizhou Province)	[17]
*Phaeoisaria filiformis*	unknown submerged wood	Thailand	[14]
*Phaeoisaria laianensis*	unknown submerged wood	China (Anhui Province)	[13]
*Phaeoisaria motuoensis*	unknown submerged wood	China (Xizang)	[51]
*Phaeoisaria obovata*	unknown submerged wood	China (Yunnan Province)	This study
*Phaeoisaria sedimenticola*	unknown submerged wood	China (Yunnan Province, Xizang)	This study [51]
*Phaeoisaria sparsa*	unknown submerged wood	Brunei, Malaysia	[30]
*Phaeoisaria synnematica*	unknown submerged wood	China (Yunnan Province)	This study
*Pleurotheciella aquatica*	unknown submerged wood	China (Yunnan Province)	[12]
*Pleurotheciella brachyspora*	unknown submerged wood	China (Yunnan Province)	This study
*Pleurotheciella centenaria*	unknown submerged wood	Canada	[15]
*Pleurotheciella erumpens*	submerged wood of various species	France, Spain	[4]
*Pleurotheciella fusiformis*	unknown submerged wood	China (Yunnan Province)	[12]
*Pleurotheciella guttulata*	unknown submerged wood	China (Yunnan Province)	[12]
*Pleurotheciella hyalospora*	unknown submerged wood	China (Yunnan Province, Guizhou Province)	This study [18]
*Pleurotheciella krabiensis*	unknown submerged wood	Thailand	[17]
*Pleurotheciella longidenticulata*	unknown submerged wood	China (Yunnan Province)	This study
*Pleurotheciella nilotica*	unknown submerged wood	China (Yunnan Province) Egypt, Thailand,	[7,61]
*Pleurotheciella rivularia*	submerged wood of *Hedera helix* and *Platanus* sp.	France	[15]
*Pleurotheciella lunata*	unknown submerged wood	China (Yunnan Province)	[12]
*Pleurotheciella obliqua*	unknown submerged wood	China (Yunnan Province)	This study
*Pleurotheciella saprophytica*	unknown submerged wood	China (Yunnan Province)	[12]
*Pleurotheciella submersa*	unknown submerged wood	China (Yunnan Province)	[12]
*Pleurotheciella sympodia*	unknown submerged wood	Thailand	[7]
*Pleurotheciella tropica*	unknown submerged wood	Thailand	[17]
*Pleurotheciella uniseptata*	unknown submerged wood	China (Yunnan Province)	[12]
*Pleurothecium aquaticum*	unknown submerged wood	China (Yunnan Province), Thailand	[8,12]
*Pleurothecium aquisubtropicum*	unknown submerged wood	China (Guizhou Province)	[9]
*Pleurothecium aseptatum*	unknown submerged wood	China (Guizhou Province)	[18]
*Pleurothecium brunius*	unknown submerged wood	Australia	[62]
*Pleurothecium floriforme*	unknown submerged wood	Thailand	[56]
*Pleurothecium guttulatum*	unknown submerged wood	China (Yunnan Province)	[7]
*Pleurothecium hainanense*	unknown submerged wood	China (Hainan Province)	[63]
*Pleurothecium pulneyense*	unknown submerged wood	China (Yunnan Province), India	[14]
*Pleurothecium recurvatum*	unknown submerged wood	China (Yunnan Province)	[14]
*Pleurothecium semifecundum*	submerged wood of *Sambucus nigra*	France	[15]
*Rhexoacrodictys erecta*	unknown submerged wood	China (Yunnan Province)	[7]
*Rhexoacrodictys fimicola*	unknown submerged wood	Thailand	[8]
*Rhexoacrodictys melanospora*	unknown submerged wood	China (Yunnan Province)	This study
*Saprodesmium dematiosporum*	unknown submerged wood	China (Yunnan Province)	[6]
*Sterigmatobotrys macrocarpa*	unknown submerged wood	Canada, UK	[14]
*Sterigmatobotrys rudis*	unknown submerged wood	China (Yunnan Province), UK	[30,60]
*Sterigmatobotrys uniseptata*	unknown submerged wood	China (Yunnan Province, Taiwan)	[14]

## Data Availability

Data are contained within the article.

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
