# Peer review of "New Species and Records of Pleurotheciaceae from Karst Landscapes in Yunnan Province, China"

_jof, 2024, doi:10.3390/jof10080516_

Round 1

Reviewer 1 Report

The reviewed MS is an example of a comprehensive, detailed, and well-organized paper dedicated to the description of several new fungal species. The obtained results  are important for mycology and ecology. The authors used proper methodology, which includes sample collection, morphological analysis, and molecular-genetic analysis. The images of fungi are excellent! The results of the study presented in the tables, high-quality figures.  The new species descriptions are very detailed. The conclusions of the paper are consistent. The reference list contains recent publications, but most of the literature is not new. Data availability statements should be explained.

I can recommend the MS for publication after some corrections.

Major comments:

  1. I recommend you extend the abstract and include in the final part the sentence about the significance of your study to fungal taxonomy, ecology, and biogeography.
  2. At the end of the article, add a final paragraph in which you can emphasize the importance of your research and the prospects for its further development.
  3. Please add about 5–6 recent publications (2019–2024) to the reference list.

Minor comments:

Lines 26–28 and further: Add authors of genus and species at the first mention.

Line 61: Try to replace the word “introduced” with a synonym.

Line 95: Add a brief description of the sample processing.

Line 347: What does PDA mean?

Author Response

Dear Reviewer: 

    Thank you for your valuable comments on our manuscript, and detailed corrections are already in our replies and re-uploaded versions.

Reviewer 2 Report

This study contributes to species diversity (new species and records) of Pleurotheciaceae from karst landscapes in Yunnan Province, China

 My only concern is that the phylogenetic tree the authors made did not take into account the different patterns of nucleotide substitutions for different regions. I would like the tree to be rebuilt or the authors to explain that this did not affect the topology.

The points below speak in favor of accepting the manuscript for the journal:

*The manuscript is clear, relevant for the field and well-structured.

*The abstract presented in the article characterizes the subject, reflects the purpose of the study, the main content and novelty of the article.

*The introduction contains historical and theoretical data according to modern literary sources.

  Nevertheless I have comments to improve the article. Please, see them below.

Reviewer comments:

Line 12 - Pleurotheciaceae is a genera-rich and highly diverse group – why not “family” or “taxon”?

Line 14, 19 and further in the text - collections – to my mind not a very appropriate term.

Line 21 and further in the text - Pleurotheciales, Savoryellomycetidae > Pleurotheciales, Savoryellomycetidaeitalicized only taxa below family only

Line 23 - Five new species >  new species. Keywords in alphabetical order.

Line 43 - Obliquifusoideum is characterized >  The genus is characterized

Line 93 – include the information about the year when the samples were collected

Line 99 and further in the text – For all equipment, indicate in brackets (manufacturer, city, country)

Line 143 – What program used for estimation of best substitution model? Moreover, the use of model partitions in the analysis would have been beneficial, given the regions' distinct mutational dynamics.

Line 153 – The dimensions of Table 1 exceed the optimal limits for this document. It should be relocated to the supplementary files.

Line 183 – Please clarify whether this is an ML phylogram.

Line 186 – To ensure greater consistency with Table S1, newly obtained sequences should be indicated by bold font, while ex-type strains should be marked with an "T."

Line 220, 223, 226 - Obliquifusoideum triseptatum > O. triseptatum

Line 234 - [1,4, 6,11,14] > [1,4,6,11,14]

Line 238, next Figs – Obliquifusoideum triseptatum not bold. All figure captions should be revised as follows: a, b Ascomata on the substratum; c vertical section of ascoma; d, e – structure of peridium; f paraphyses; g–j asci; K apex of ascus; l–o ascospores; p – germinated ascospore; q, r – colony on PDA from surface and reverse. Scale bars: c = 40 μm, d, k–p = 10 μm, e–j = 20 μm.

Line 275 – Phaeoisaria obovata > P. obovata

Lines 318-320 – Phaeoisaria sedimenticola > P. sedimenticola

Line 417 - Pleurotheciella brachyspora > P. brachyspora

Lines 457, 460 – Pleurotheciella hyalospora > P. hyalospora

Lines 503-510 – Pleurotheciella longidenticulata > P. longidenticulata

Lines 562-571 – Pleurotheciella obliqua > P. obliqua

Line 661 – The dimensions of Table 2 exceed the optimal limits for this document. It should be relocated to the supplementary files.

Author Response

(The authors gave the same response as above.)
